# Variance-based Regularization with Convex Objectives

**Hongseok Namkoong**
Stanford University
hnamk@stanford.edu

**John C. Duchi**
Stanford University
jduchi@stanford.edu

## Abstract

We develop an approach to risk minimization and stochastic optimization that provides a convex surrogate for variance, allowing near-optimal and computationally efficient trading between approximation and estimation error. Our approach builds off of techniques for distributionally robust optimization and Owen's empirical likelihood, and we provide a number of finite-sample and asymptotic results characterizing the theoretical performance of the estimator. In particular, we show that our procedure comes with certificates of optimality, achieving (in some scenarios) faster rates of convergence than empirical risk minimization by virtue of automatically balancing bias and variance. We give corroborating empirical evidence showing that in practice, the estimator indeed trades between variance and absolute performance on a training sample, improving out-of-sample (test) performance over standard empirical risk minimization for a number of classification problems.

## 1 Introduction

Let $\mathcal{X}$ be a sample space, $P_0$ a distribution on $\mathcal{X}$, and $\Theta$ a parameter space. For a loss function $\ell : \Theta \times \mathcal{X} \to \mathbb{R}$, consider the problem of finding $\theta \in \Theta$ minimizing the risk

$$R(\theta) := \mathbb{E}[\ell(\theta, X)] = \int \ell(\theta, x) dP(x) \tag{1}$$

given a sample $\{X_1, \ldots, X_n\}$ drawn i.i.d. according to the distribution $P$. Under appropriate conditions on the loss $\ell$, parameter space $\Theta$, and random variables $X$, a number of researchers [2, 6, 12, 7, 3] have shown results of the form that with high probability,

$$R(\theta) \le \frac{1}{n} \sum_{i=1}^{n} \ell(\theta, X_i) + C_1 \sqrt{\frac{\mathrm{Var}(\ell(\theta, X))}{n}} + \frac{C_2}{n} \ \text{ for all } \theta \in \Theta \tag{2}$$

where $C_1$ and $C_2$ depend on the parameters of problem (1) and the desired confidence guarantee. Such bounds justify empirical risk minimization, which chooses $\widehat{\theta}_n$ to minimize $\frac{1}{n} \sum_{i=1}^{n} \ell(\theta, X_i)$ over $\theta \in \Theta$. Further, these bounds showcase a tradeoff between bias and variance, where we identify the bias (or approximation error) with the empirical risk $\frac{1}{n} \sum_{i=1}^{n} \ell(\theta, X_i)$, while the variance arises from the second term in the bound.

Considering the bias-variance tradeoff (1) in statistical learning, it is natural to instead choose $\theta$ to directly minimize a quantity trading between approximation and estimation error:

$$\frac{1}{n} \sum_{i=1}^{n} \ell(\theta, X_i) + C \sqrt{\frac{\mathrm{Var}_{\widehat{P}_n}(\ell(\theta, X))}{n}}, \tag{3}$$

where $\mathrm{Var}_{\widehat{P}_n}$ denotes the empirical variance. Maurer and Pontil [16] consider this idea, giving guarantees on the convergence and good performance of such a procedure. Unfortunately, even when

the loss $\ell$ is convex in $\theta$, the formulation (3) is generally non-convex, which limits the applicability of procedures that minimize the variance-corrected empirical risk (3). In this paper, we develop an approach based on Owen's empirical likelihood [19] and ideas from distributionally robust optimization [4, 5, 10] that—whenever the loss $\ell$ is convex—provides a tractable *convex* formulation closely approximating the penalized risk (3). We give a number of theoretical guarantees and empirical evidence for its performance.

To describe our approach, we require a few definitions. For a convex function $\phi : \mathbb{R}_+ \to \mathbb{R}$ with $\phi(1) = 0$, $D_\phi(P\|Q) = \int_{\mathcal{X}} \phi(\frac{dP}{dQ})dQ$ is the $\phi$-*divergence* between distributions $P$ and $Q$ defined on $\mathcal{X}$. Throughout this paper, we use $\phi(t) = \frac{1}{2}(t-1)^2$, which gives the $\chi^2$-divergence. Given $\phi$ and an i.i.d. sample $X_1, \ldots, X_n$, we define the $\rho$-*neighborhood of the empirical distribution*

$$\mathcal{P}_n := \left\{ \text{distributions } P \text{ s.t. } D_\phi(P\|\widehat{P}_n) \leq \frac{\rho}{n} \right\},$$

where $\widehat{P}_n$ denotes the empirical distribution of the sample $\{X_i\}_{i=1}^n$, and our choice $\phi(t) = \frac{1}{2}(t-1)^2$ means that $\mathcal{P}_n$ has support $\{X_i\}_{i=1}^n$. We then define the *robustly regularized risk*

$$R_n(\theta, \mathcal{P}_n) := \sup_{P \in \mathcal{P}_n} \mathbb{E}_P[\ell(\theta, X)] = \sup_P \left\{ \mathbb{E}_P[\ell(\theta, X)] : D_\phi(P\|\widehat{P}_n) \leq \frac{\rho}{n} \right\}. \tag{4}$$

As it is the supremum of a family of convex functions, the robust risk $\theta \mapsto R_n(\theta, \mathcal{P}_n)$ is convex in $\theta$ regardless of the value of $\rho \geq 0$ whenever the original loss $\ell(\cdot; X)$ is convex and $\Theta$ is a convex set. Namkoong and Duchi [18] propose a stochastic procedure for minimizing (4) almost as fast as stochastic gradient descent. See Appendix C for a detailed account of an alternative method.

We show that the robust risk (4) provides an excellent surrogate for the variance-regularized quantity (3) in a number of ways. Our first result (Thm. 1 in Sec. 2) is that for bounded loss functions,

$$R_n(\theta, \mathcal{P}_n) = \mathbb{E}_{\widehat{P}_n}[\ell(\theta, X)] + \sqrt{\frac{2\rho}{n}\mathrm{Var}_{\widehat{P}_n}(\ell(\theta, X))} + \varepsilon_n(\theta), \tag{5}$$

where $\varepsilon_n(\theta) \leq 0$ and is $O(1/n)$ uniformly in $\theta$. We show that when $\ell(\theta, X)$ has suitably large variance, we have $\varepsilon_n = 0$ with high probability. With the expansion (5) in hand, we can show a number of finite-sample convergence guarantees for the robustly regularized estimator

$$\widehat{\theta}_n^{\mathrm{rob}} \in \operatorname*{argmin}_{\theta \in \Theta} \left\{ \sup_P \left\{ \mathbb{E}_P[\ell(\theta, X)] : D_\phi(P\|\widehat{P}_n) \leq \frac{\rho}{n} \right\} \right\}. \tag{6}$$

Based on the expansion (5), solutions $\widehat{\theta}_n^{\mathrm{rob}}$ of problem (6) enjoy automatic finite sample optimality certificates: for $\rho \geq 0$, with probability at least $1 - C_1 \exp(-\rho)$ we have

$$\mathbb{E}[\ell(\widehat{\theta}_n^{\mathrm{rob}}; X)] \leq R_n(\widehat{\theta}_n^{\mathrm{rob}}; \mathcal{P}_n) + \frac{C_2\rho}{n} = \inf_{\theta \in \Theta} R_n(\theta, \mathcal{P}_n) + \frac{C_2\rho}{n}$$

where $C_1, C_2$ are constants (which we specify) that depend on the loss $\ell$ and domain $\Theta$. That is, with high probability the robust solution has risk no worse than the optimal finite sample robust objective up to an $O(\rho/n)$ error term. To guarantee a desired level of risk performance with probability $1 - \delta$, we may specify the robustness penalty $\rho = O(\log \frac{1}{\delta})$.

Secondly, we show that the procedure (6) allows us to automatically and near-optimally trade between approximation and estimation error (bias and variance), so that

$$\mathbb{E}[\ell(\widehat{\theta}_n^{\mathrm{rob}}; X)] \leq \inf_{\theta \in \Theta} \left\{ \mathbb{E}[\ell(\theta; X)] + 2\sqrt{\frac{2\rho}{n}\mathrm{Var}(\ell(\theta; X))} \right\} + \frac{C\rho}{n}$$

with high probability. When there are parameters $\theta$ with small risk $R(\theta)$ (relative to the optimal parameter $\theta^\star$) and small variance $\mathrm{Var}(\ell(\theta, X))$, this guarantees that the excess risk $R(\widehat{\theta}_n^{\mathrm{rob}}) - R(\theta^\star)$ is essentially of order $O(\rho/n)$, where $\rho$ governs our desired confidence level. We give an explicit example in Section 3.2 where our robustly regularized procedure (6) converges at $O(\log n/n)$ compared to $O(1/\sqrt{n})$ of empirical risk minimization.

Bounds that trade between risk and variance are known in a number of cases in the empirical risk minimization literature [15, 22, 2, 1, 6, 3, 7, 12], which is relevant when one wishes to achieve "fast

rates" of convergence for statistical learning algorithms. In many cases, such tradeoffs require either conditions such as the Mammen-Tsybakov noise condition [15, 6] or localization results [3, 2, 17] made possible by curvature conditions that relate the risk and variance. The robust solutions (6) enjoy a variance-risk tradeoff that is differen but holds essentially without conditions except compactness of $\Theta$. We show in Section 3.3 that the robust solutions enjoy fast rates of convergence under typitcal curvature conditions on the risk $R$.

We complement our theoretical results in Section 4, where we conclude by providing two experiments comparing empirical risk minimization (ERM) strategies to robustly-regularized risk minimization (6). These results validate our theoretical predictions, showing that the robust solutions are a practical alternative to empirical risk minimization. In particular, we observe that the robust solutions outperform their ERM counterparts on "harder" instances with higher variance. In classification problems, for example, the robustly regularized estimators exhibit an interesting tradeoff, where they improve performance on rare classes (where ERM usually sacrifices performance to improve the common cases—increasing variance slightly) at minor cost in performance on common classes.

## 2   Variance Expansion

We begin our study of the robust regularized empirical risk $R_n(\theta, \mathcal{P}_n)$ by showing that it is a good approximation to the empirical risk plus a variance term (5). Although the variance of the loss is in general non-convex, the robust formulation (6) is a convex optimization problem for variance regularization whenever the loss function is convex [cf. 11, Prop. 2.1.2.].

To gain intuition for the variance expansion that follows, we consider the following equivalent formulation for the robust objective $\sup_{P \in \mathcal{P}_n} \mathbb{E}_P[Z]$

$$\underset{p}{\text{maximize}} \ \sum_{i=1}^{n} p_i z_i \ \text{ subject to } p \in \mathcal{P}_n = \left\{ p \in \mathbb{R}_+^n : \frac{1}{2} \|np - \mathbf{1}\|_2^2 \leq \rho, \langle \mathbf{1}, p \rangle = 1 \right\}, \quad (7)$$

where $z \in \mathbb{R}^n$ is a vector. For simplicity, let $s_n^2 = \frac{1}{n} \|z\|_2^2 - (\overline{z})^2 = \frac{1}{n} \|z - \overline{z}\|_2^2$ denote the empirical "variance" of the vector $z$, where $\overline{z} = \frac{1}{n} \langle \mathbf{1}, z \rangle$ is the mean value of $z$. Then by introducing the variable $u = p - \frac{1}{n}\mathbf{1}$, the objective in problem (7) satisfies $\langle p, z \rangle = \overline{z} + \langle u, z \rangle = \overline{z} + \langle u, z - \overline{z} \rangle$ because $\langle u, \mathbf{1} \rangle = 0$. Thus problem (7) is equivalent to solving

$$\underset{u \in \mathbb{R}^n}{\text{maximize}} \ \overline{z} + \langle u, z - \overline{z} \rangle \ \text{ subject to } \|u\|_2^2 \leq \frac{2\rho}{n^2}, \ \langle \mathbf{1}, u \rangle = 0, \ u \geq -\frac{1}{n}.$$

Notably, by the Cauchy-Schwarz inequality, we have $\langle u, z - \overline{z} \rangle \leq \sqrt{2\rho} \|z - \overline{z}\|_2 / n = \sqrt{2\rho s_n^2 / n}$, and equality is attained if and only if

$$u_i = \frac{\sqrt{2\rho}(z_i - \overline{z})}{n \|z - \overline{z}\|_2} = \frac{\sqrt{2\rho}(z_i - \overline{z})}{n \sqrt{n s_n^2}}.$$

Of course, it is possible to choose such $u_i$ while satisfying the constraint $u_i \geq -1/n$ if and only if

$$\min_{i \in [n]} \frac{\sqrt{2\rho}(z_i - \overline{z})}{\sqrt{n s_n^2}} \geq -1. \quad (8)$$

Thus, if inequality (8) holds for the vector $z$—that is, there is enough variance in $z$—we have

$$\sup_{p \in \mathcal{P}_n} \langle p, z \rangle = \overline{z} + \sqrt{\frac{2\rho s_n^2}{n}}.$$

For losses $\ell(\theta, X)$ with enough variance relative to $\ell(\theta, X_i) - \mathbb{E}_{\widehat{P}_n}[\ell(\theta, X_i)]$, that is, those satisfying inequality (8), then, we have

$$R_n(\theta, \mathcal{P}_n) = \mathbb{E}_{\widehat{P}_n}[\ell(\theta, X)] + \sqrt{\frac{2\rho}{n} \text{Var}_{\widehat{P}_n}(\ell(\theta, X))}.$$

A slight elaboration of this argument, coupled with the application of a few concentration inequalities, yields the next theorem. Recall that $\phi(t) = \frac{1}{2}(t-1)^2$ in our definition of the $\phi$-divergence.

**Theorem 1.** *Let $Z$ be a random variable taking values in $[M_0, M_1]$ where $M = M_1 - M_0$ and fix $\rho \geq 0$. Then*

$$\left( \sqrt{\frac{2\rho}{n} \mathrm{Var}_{\widehat{P}_n}(Z)} - \frac{2M\rho}{n} \right)_+ \leq \sup_P \left\{ \mathbb{E}_P[Z] : D_\phi(P \| \widehat{P}_n) \leq \frac{\rho}{n} \right\} - \mathbb{E}_{\widehat{P}_n}[Z] \leq \sqrt{\frac{2\rho}{n} \mathrm{Var}_{\widehat{P}_n}(Z)}.$$

$$(9)$$

*If $n \geq \max\{\frac{24\rho}{\mathrm{Var}(Z)}, \frac{16}{\mathrm{Var}(Z)}, 1\} M^2$ and we set $t_n = \sqrt{\mathrm{Var}(Z)} \left( \sqrt{1 - n^{-1}} - \frac{1}{2} \right) - \frac{M^2}{n} \geq \sqrt{\frac{\mathrm{Var}(Z)}{18}}$,*

$$\sup_{P : D_\phi(P \| \widehat{P}_n) \leq \frac{\rho}{n}} \mathbb{E}_P[Z] = \mathbb{E}_{\widehat{P}_n}[Z] + \sqrt{\frac{2\rho}{n} \mathrm{Var}_{\widehat{P}_n}(Z)} \tag{10}$$

*with probability at least $1 - \exp(-\frac{n t_n^2}{2M^2}) \geq 1 - \exp(-\frac{n \mathrm{Var}(Z)}{36 M^2})$.*

See Appendix A.1 for the proof. Inequality (9) and the exact expansion (10) show that, at least for bounded loss functions $\ell$, the robustly regularized risk (4) is a natural (and convex) surrogate for empirical risk plus standard deviation of the loss, and the robust formulation approximates exact variance regularization with a convex penalty.

We also provide a uniform variant of Theorem 1 based on the standard notion of the covering number, which we now define. Let $\mathcal{V}$ be a vector space with (semi)norm $\|\cdot\|$ on $\mathcal{V}$, and let $V \subset \mathcal{V}$. We say a collection $v_1, \ldots, v_N \subset V$ is an $\epsilon$-*cover* of $V$ if for each $v \in V$, there exists $v_i$ such that $\|v - v_i\| \leq \epsilon$. The *covering number* of $V$ with respect to $\|\cdot\|$ is then $N(V, \epsilon, \|\cdot\|) := \inf\{N \in \mathbb{N} : \text{ there is an } \epsilon\text{-cover of } V \text{ with respect to } \|\cdot\|\}$. Now, let $\mathcal{F}$ be a collection of functions $f : \mathcal{X} \to \mathbb{R}$, and define the $L^\infty(\mathcal{X})$-norm by $\|f - g\|_{L^\infty(\mathcal{X})} := \sup_{x \in \mathcal{X}} |f(x) - g(x)|$. Although we state our results abstractly, we typically take $\mathcal{F} := \{\ell(\theta, \cdot) \mid \theta \in \Theta\}$.

As a motivating example, we give the following standard bound on the covering number of Lipschitz losses [24].

**Example 1:** Let $\Theta \subset \mathbb{R}^d$ and assume that $\ell : \Theta \times \mathcal{X} \to \mathbb{R}$ is $L$-Lipschitz in $\theta$ with respect to the $\ell_2$-norm for all $x \in \mathcal{X}$, meaning that $|\ell(\theta, x) - \ell(\theta', x)| \leq L \|\theta - \theta'\|_2$. Then taking $\mathcal{F} = \{\ell(\theta, \cdot) : \theta \in \Theta\}$, any $\epsilon$-covering $\{\theta_1, \ldots, \theta_N\}$ of $\Theta$ in $\ell_2$-norm guarantees that $\min_i |\ell(\theta, x) - \ell(\theta_i, x)| \leq L\epsilon$ for all $\theta, x$. That is,

$$N(\mathcal{F}, \epsilon, \|\cdot\|_{L^\infty(\mathcal{X})}) \leq N(\Theta, \epsilon/L, \|\cdot\|_2) \leq \left( 1 + \frac{\mathrm{diam}(\Theta) L}{\epsilon} \right)^d,$$

where $\mathrm{diam}(\Theta) = \sup_{\theta, \theta' \in \Theta} \|\theta - \theta'\|_2$. Thus $\ell_2$-covering numbers of $\Theta$ control $L^\infty$-covering numbers of the family $\mathcal{F}$. ♦

With this definition, we provide a result showing that the variance expansion (5) holds uniformly for all functions with *enough* variance.

**Theorem 2.** *Let $\mathcal{F}$ be a collection of bounded functions $f : \mathcal{X} \to [M_0, M_1]$ where $M = M_1 - M_0$, and let $\tau \geq 0$ be a constant. Define $\mathcal{F}_{\geq \tau} := \{f \in \mathcal{F} : \mathrm{Var}(f) \geq \tau^2\}$ and $t_n = \tau(\sqrt{1 - n^{-1}} - \frac{1}{2}) - \frac{M^2}{n}$. If $\tau^2 \geq \frac{32\rho M^2}{n}$, then with probability at least $1 - N\left(\mathcal{F}, \frac{\tau}{32}, \|\cdot\|_{L^\infty(\mathcal{X})}\right) \exp\left(-\frac{n t_n^2}{2M^2}\right)$, we have for all $f \in \mathcal{F}_{\geq \tau}$*

$$\sup_{P : D_\phi(P \| \widehat{P}_n) \leq \frac{\rho}{n}} \mathbb{E}_P[f(X)] = \mathbb{E}_{\widehat{P}_n}[f(X)] + \sqrt{\frac{2\rho}{n} \mathrm{Var}_{\widehat{P}_n}(f(X))}. \tag{11}$$

We prove the theorem in Section A.2. Theorem 2 shows that the variance expansion of Theorem 1 holds uniformly for all functions $f$ with sufficient variance. See Duchi, Glynn, and Namkoong [10] for an asymptotic analogue of the equality (11) for heavier tailed random variables.

## 3 Optimization by Minimizing the Robust Loss

Based on the variance expansions in the preceding section, we show that the robust solution (6) automatically trades between approximation and estimation error. In addition to $\|\cdot\|_{L^\infty(\mathcal{X})}$-covering

numbers defined in the previous section, we use the tighter notion of empirical $\ell_\infty$-covering numbers. For $x \in \mathcal{X}^n$, define $\mathcal{F}(x) = \{(f(x_1), \ldots, f(x_n)) : f \in \mathcal{F}\}$ and the empirical $\ell_\infty$-covering numbers $N_\infty(\mathcal{F}, \epsilon, n) := \sup_{x \in \mathcal{X}^n} N(\mathcal{F}(x), \epsilon, \|\cdot\|_\infty)$, which bound the number of $\ell_\infty$-balls of radius $\epsilon$ required to cover $\mathcal{F}(x)$. Note that we always have $N_\infty(\mathcal{F}) \leq N(\mathcal{F})$.

Typically, we consider the function class $\mathcal{F} := \{\ell(\theta, \cdot) : \theta \in \Theta\}$, though we state our minimization results abstractly. Although the below result is in terms of covering numbers for ease of exposition, a variant holds depending on localized Rademacher averages [2] of the class $\mathcal{F}$, which can yield tighter guarantees (we omit such results for lack of space). We prove the following theorem in Section A.3.

**Theorem 3.** *Let $\mathcal{F}$ be a collection of functions $f : \mathcal{X} \to [M_0, M_1]$ with $M = M_1 - M_0$. Define the empirical minimizer*

$$\widehat{f} \in \operatorname*{argmin}_{f \in \mathcal{F}} \left\{ \sup_P \left\{ \mathbb{E}_P[f(X)] : D_\phi(P\|\widehat{P}_n) \leq \frac{\rho}{n} \right\} \right\}.$$

*Then for $\rho \geq t$, with probability at least $1 - 2(N(\mathcal{F}, \epsilon, \|\cdot\|_{L^\infty(\mathcal{X})}) + 1)e^{-t}$,*

$$\mathbb{E}[\widehat{f}(X)] \leq \sup_{P : D_\phi(P\|\widehat{P}_n) \leq \frac{\rho}{n}} \mathbb{E}_P[\widehat{f}(X)] + \frac{7M\rho}{n} + \left(2 + \sqrt{\frac{2t}{n-1}}\right)\epsilon \tag{12a}$$

$$\leq \inf_{f \in \mathcal{F}} \left\{ \mathbb{E}[f] + 2\sqrt{\frac{2\rho}{n}\mathrm{Var}(f)} \right\} + \frac{11M\rho}{n} + \left(2 + \sqrt{\frac{2t}{n-1}}\right)\epsilon. \tag{12b}$$

*Further, for $n \geq \frac{8M^2}{t}$, $t \geq \log 12$, and $\rho \geq 9t$, with probability at least $1 - 2(3N_\infty(\mathcal{F}, \epsilon, 2n) + 1)e^{-t}$,*

$$\mathbb{E}[\widehat{f}(X)] \leq \sup_{P : D_\phi(P\|\widehat{P}_n) \leq \frac{\rho}{n}} \mathbb{E}_P[\widehat{f}(X)] + \frac{11}{3}\frac{M\rho}{n} + \left(2 + 4\sqrt{\frac{2t}{n}}\right)\epsilon \tag{13a}$$

$$\leq \inf_{f \in \mathcal{F}} \left\{ \mathbb{E}[f] + 2\sqrt{\frac{2\rho}{n}\mathrm{Var}(f)} \right\} + \frac{19M\rho}{3n} + \left(2 + 4\sqrt{\frac{2t}{n}}\right)\epsilon. \tag{13b}$$

Unlike analogous results for empirical risk minimization [6], Theorem 3 does not require the self-bounding type assumption $\mathrm{Var}(f) \leq B\mathbb{E}[f]$. A consequence of this is that when $v = \mathrm{Var}(f^*)$ is small, where $f^* \in \operatorname{argmin}_{f \in \mathcal{F}} \mathbb{E}[f]$, we achieve $O(1/n + \sqrt{v/n})$ (fast) rates of convergence. This condition is different from the typical conditions required for empirical risk minimization to have fast rates of convergence, highlighting the possibilities of variance-based regularization. It will be interesting to understand appropriate low-noise conditions (e.g. the Mammen-Tsybakov noise condition [15, 6]) guaranteeing good performance. Additionally, the robust objective $R_n(\theta, \mathcal{P}_n)$ is an empirical likelihood confidence bound on the population risk [10], and as empirical likelihood confidence bounds are self-normalizing [19], other fast-rate generalizations may exist.

## 3.1 Consequences of Theorem 3

We now turn to a number of corollaries that expand on Theorem 3 to investigate its consequences. Our first corollary shows that Theorem 3 applies to standard Vapnik-Chervonenkis (VC) classes. As VC dimension is preserved through composition, this result also extends to the procedure (6) in typical empirical risk minimization scenarios. See Section A.4 for its proof.

**Corollary 3.1.** *In addition to the conditions of Theorem 3, let $\mathcal{F}$ have finite VC-dimension $\mathsf{VC}(\mathcal{F})$. Then for a numerical constant $c < \infty$, the bounds* (13) *hold with probability at least* $1 - \left(c\,\mathsf{VC}(\mathcal{F})\left(\frac{16Mne}{\epsilon}\right)^{\mathsf{VC}(\mathcal{F})-1} + 2\right)e^{-t}$.

Next, we focus more explicitly on the estimator $\widehat{\theta}_n^{\mathrm{rob}}$ defined by minimizing the robust regularized risk (6). Let us assume that $\Theta \subset \mathbb{R}^d$, and that we have a typical linear modeling situation, where a loss $h$ is applied to an inner product, that is, $\ell(\theta, x) = h(\theta^\top x)$. In this case, by making the substitution that the class $\mathcal{F} = \{\ell(\theta, \cdot) : \theta \in \Theta\}$ in Corollary 3.1, we have $\mathsf{VC}(\mathcal{F}) \leq d$, and we obtain the following corollary. Recall the definition (1) of the population risk $R(\theta) = \mathbb{E}[\ell(\theta, X)]$, and the uncertainty set $\mathcal{P}_n = \{P : D_\phi(P\|\widehat{P}_n) \leq \frac{\rho}{n}\}$, and that $R_n(\theta, \mathcal{P}_n) = \sup_{P \in \mathcal{P}_n} \mathbb{E}_P[\ell(\theta, X)]$. By setting $\epsilon = M/n$ in Corollary 3.1, we obtain the following result.

**Corollary 3.2.** *Let the conditions of the previous paragraph hold and assume that $\ell(\theta, x) \in [0, M]$ for all $\theta \in \Theta, x \in \mathcal{X}$. Then if $n \geq \rho \geq 9 \log 12$,*

$$R(\widehat{\theta}_n^{\mathrm{rob}}) \leq R_n(\widehat{\theta}_n^{\mathrm{rob}}, \mathcal{P}_n) + \frac{11M\rho}{3n} + \frac{4M}{n} \leq \inf_{\theta \in \Theta} \left\{ R(\theta) + 2\sqrt{\frac{2\rho}{n} \mathrm{Var}(\ell(\theta; X))} \right\} + \frac{11M\rho}{n}$$

*with probability at least $1 - 2\exp(c_1 d \log n - c_2 \rho)$, where $c_i$ are universal constants with $c_2 \geq 1/9$.*

Unpacking Theorem 3 and Corollary 3.2 a bit, the first result (13a) provides a high-probability guarantees that the true expectation $\mathbb{E}[\widehat{f}]$ cannot be more than $O(1/n)$ worse than its robustly-regularized empirical counterpart, that is, $R(\widehat{\theta}_n^{\mathrm{rob}}) \leq R_n(\widehat{\theta}_n^{\mathrm{rob}}, \mathcal{P}_n) + O(\rho/n)$, which is (roughly) a consequence of uniform variants of Bernstein's inequality. The second result (13b) guarantee the convergence of the empirical minimizer to a parameter with risk at most $O(1/n)$ larger than the best possible variance-corrected risk. In the case that the losses take values in $[0, M]$, then $\mathrm{Var}(\ell(\theta, X)) \leq MR(\theta)$, and thus for $\epsilon = 1/n$ in Theorem 3, we obtain

$$R(\widehat{\theta}_n^{\mathrm{rob}}) \leq R(\theta^\star) + C\sqrt{\frac{M\rho R(\theta^\star)}{n}} + C\frac{M\rho}{n},$$

a type of result well-known and achieved by empirical risk minimization for bounded nonnegative losses [6, 26, 25]. In some scenarios, however, the variance may satisfy $\mathrm{Var}(\ell(\theta, X)) \ll MR(\theta)$, yielding improvements.

To give an alternative variant of Corollary 3.2, let $\Theta \subset \mathbb{R}^d$ and assume that for each $x \in \mathcal{X}$, $\inf_{\theta \in \Theta} \ell(\theta, x) = 0$ and that $\ell$ is $L$-Lipschitz in $\theta$. If $D := \mathrm{diam}(\Theta) = \sup_{\theta, \theta' \in \Theta} \|\theta - \theta'\| < \infty$, then $0 \leq \ell(\theta, x) \leq L \mathrm{diam}(\Theta) =: M$.

**Corollary 3.3.** *Let the conditions of the preceeding paragraph hold. Set $t = \rho = \log 2n + d \log(2nDL)$ and $\epsilon = \frac{1}{n}$ in Theorem 3 and assume that $D \lesssim n^k$ and $L \lesssim n^k$ for a numerical constant $k$. With probability at least $1 - 1/n$,*

$$\mathbb{E}[\ell(\widehat{\theta}_n^{\mathrm{rob}}; X)] = R(\widehat{\theta}_n^{\mathrm{rob}}) \leq \inf_{\theta \in \Theta} \left\{ R(\theta) + C\sqrt{\frac{d \mathrm{Var}(\ell(\theta, X))}{n} \log n} \right\} + C\frac{dLD \log n}{n}$$

*where $C$ is a numerical constant.*

### 3.2 Beating empirical risk minimization

We now provide an example in which the robustly-regularized estimator (6) exhibits a substantial improvement over empirical risk minimization. We expect the robust approach to offer performance benefits in situations in which the empirical risk minimizer is highly sensitive to noise, say, because the losses are piecewise linear, and slight under- or over-estimates of slope may significantly degrade solution quality. With this in mind, we construct a toy 1-dimensional example—estimating the median of a distribution supported on $\mathcal{X} = \{-1, 0, 1\}$—in which the robust-regularized estimator has convergence rate $\log n/n$, while empirical risk minimization is at best $1/\sqrt{n}$.

Define the loss $\ell(\theta; x) = |\theta - x| - |x|$, and for $\delta \in (0, 1)$ let the distribution $P$ be defined by $P(X = 1) = \frac{1-\delta}{2}$, $P(X = -1) = \frac{1-\delta}{2}$, $P(X = 0) = \delta$. Then for $\theta \in \mathbb{R}$, the risk of the loss is

$$R(\theta) = \delta|\theta| + \frac{1-\delta}{2}|\theta - 1| + \frac{1-\delta}{2}|\theta + 1| - (1 - \delta).$$

By symmetry, it is clear that $\theta^\star := \mathrm{argmin}_\theta R(\theta) = 0$, which satisfies $R(\theta^\star) = 0$. (Note that $\ell(\theta, x) = \ell(\theta, x) - \ell(\theta^\star, x)$.) Without loss of generality, we assume that $\Theta = [-1, 1]$. Define the empirical risk minimizer and the robust solution

$$\widehat{\theta}^{\mathrm{erm}} := \mathrm{argmin}_{\theta \in \mathbb{R}} \mathbb{E}_{\widehat{P}_n}[\ell(\theta, X)] = \mathrm{argmin}_{\theta \in [-1, 1]} \mathbb{E}_{\widehat{P}_n}[|\theta - X|], \quad \widehat{\theta}_n^{\mathrm{rob}} \in \mathrm{argmin}_{\theta \in \Theta} R_n(\theta, \mathcal{P}_n).$$

Intuitively, if too many of the observations satisfy $X_i = 1$ or too many satisfy $X_i = -1$, then $\widehat{\theta}^{\mathrm{erm}}$ will be either 1 or $-1$; for small $\delta$, such events become reasonably probable. On the other hand, we have $\ell(\theta^\star; x) = 0$ for all $x \in \mathcal{X}$, so that $\mathrm{Var}(\ell(\theta^\star; X)) = 0$ and variance regularization achieves the rate $O(\log n/n)$ as opposed to empirical risk minimizer's $O(1/\sqrt{n})$. See Section A.6 for the proof.

**Proposition 1.** *Under the conditions of the previous paragraph, for $n \geq \rho = 3 \log n$, with probability at least $1 - \frac{4}{n}$, we have $R(\widehat{\theta}_n^{\mathrm{rob}}) - R(\theta^\star) \leq \frac{45 \log n}{n}$. However, with probability at least $2\Phi(-\sqrt{\frac{n}{n-1}}) - 2\sqrt{2}/\sqrt{\pi e n} \geq 2\Phi(-\sqrt{\frac{n}{n-1}}) - n^{-\frac{1}{2}}$, we have $R(\widehat{\theta}^{\mathrm{erm}}) \geq R(\theta^\star) + n^{-\frac{1}{2}}$.*

For $n \geq 20$, the probability of the latter event is $\geq .088$. Hence, for this (specially constructed) example, we see that there is a gap of nearly $n^{\frac{1}{2}}$ in order of convergence.

### 3.3 Fast Rates

In cases in which the risk $R$ has curvature, empirical risk minimization often enjoys faster rates of convergence [6, 21]. The robust solution $\widehat{\theta}_n^{\mathrm{rob}}$ similarly attains faster rates of convergence in such cases, even with approximate minimizers of $R_n(\theta, \mathcal{P}_n)$. For the risk $R$ and $\epsilon \geq 0$, let $S^\epsilon := \{\theta \in \Theta : R(\theta) \leq \inf_{\theta^\star \in \Theta} R(\theta^\star) + \epsilon\}$ denote the $\epsilon$-sub-optimal (solution) set, and similarly let $\widehat{S}^\epsilon := \{\theta \in \Theta : R_n(\theta, \mathcal{P}_n) \leq \inf_{\theta' \in \Theta} R_n(\theta', \mathcal{P}_n) + \epsilon\}$. For a vector $\theta \in \Theta$, let $\pi_S(\theta) = \operatorname{argmin}_{\theta^\star \in S} \|\theta^\star - \theta\|_2$ denote the Euclidean projection of $\theta$ onto the set $S$.

Our below result depends on a local notion of Rademacher complexity. For i.i.d. random signs $\varepsilon_i \in \{\pm 1\}$, the empirical Rademacher complexity of a function class $\mathcal{F} \subset \{f : \mathcal{X} \to \mathbb{R}\}$ is

$$\mathfrak{R}_n \mathcal{F} := \mathbb{E}\left[ \sup_{f \in \mathcal{F}} \frac{1}{n} \sum_{i=1}^{n} \varepsilon_i f(X_i) \mid X \right].$$

Although we state our results abstractly, we typically take $\mathcal{F} := \{\ell(\theta, \cdot) \mid \theta \in \Theta\}$. For example, when $\mathcal{F}$ is a VC-class, we typically have $\mathbb{E}[\mathfrak{R}_n \mathcal{F}] \lesssim \sqrt{\mathsf{VC}(\mathcal{F})/n}$. Many other bounds on $\mathbb{E}[\mathfrak{R}_n \mathcal{F}]$ are possible [1, 24, Ch. 2]. For $A \subset \Theta$ let $\mathfrak{R}_n(A)$ denote the Rademacher complexity of the localized process $\{x \mapsto \ell(\theta; x) - \ell(\pi_S(\theta); x) : \theta \in A\}$. We then have the following result, whose proof we provide in Section A.7.

**Theorem 4.** *Let $\Theta \subset \mathbb{R}^d$ be convex and let $\ell(\cdot; x)$ be convex and $L$-Lipshitz for all $x \in \mathcal{X}$. For constants $\lambda > 0$, $\gamma > 1$, and $r > 0$, assume that $R$ satisfies*

$$R(\theta) - \inf_{\theta \in \Theta} R(\theta) \geq \lambda \operatorname{dist}(\theta, S)^\gamma \ \text{ for all } \theta \text{ such that } \operatorname{dist}(\theta, S) \leq r. \tag{14}$$

*Let $t > 0$. If $0 \leq \epsilon \leq \frac{1}{2}\lambda r^\gamma$ satisfies*

$$\epsilon \geq \left(\frac{8L^2\rho}{n}\right)^{\frac{\gamma}{2(\gamma-1)}} \left(\frac{2}{\lambda}\right)^{\frac{1}{\gamma-1}} \ \text{ and } \ \frac{\epsilon}{2} \geq 2\mathbb{E}[\mathfrak{R}_n(S^{2\epsilon})] + L\left(\frac{2\epsilon}{\lambda}\right)^{\frac{1}{\gamma}} \sqrt{\frac{2t}{n}}, \tag{15}$$

*then $\mathbb{P}(\widehat{S}^\epsilon \subset S^{2\epsilon}) \geq 1 - e^{-t}$, and inequality (15) holds for all $\epsilon \gtrsim (\frac{L^2(t+\rho+d)}{\lambda^{2/\gamma}n})^{\frac{\gamma}{2(\gamma-1)}}$.*

## 4 Experiments

We present two real classification experiments to carefully compare standard empirical risk minimization (ERM) to the variance-regularized approach we present. Empirically, we show that the ERM estimator $\widehat{\theta}^{\mathrm{erm}}$ performs poorly on rare classes with (relatively) more variance, where the robust solution achieves improved classification performance on rare instances. In all our experiments, this occurs with little expense over the more common instances.

### 4.1 Protease cleavage experiments

For our first experiment, we compare our robust regularization procedure to other regularizers using the HIV-1 protease cleavage dataset from the UCI ML-repository [14]. In this binary classification task, one is given a string of amino acids (a protein) and a featurized representation of the string of dimension $d = 50960$, and the goal is to predict whether the HIV-1 virus will cleave the amino acid sequence in its central position. We have a sample of $n = 6590$ observations of this process, where the class labels are somewhat skewed: there are 1360 examples with label $Y = +1$ (HIV-1 cleaves) and 5230 examples with $Y = -1$ (does not cleave).

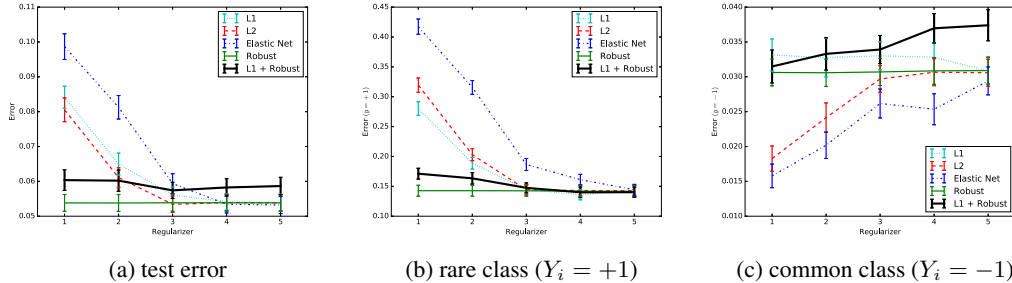

|                       |                       |                       |
|:---------------------:|:---------------------:|:---------------------:|
| (a) test error | (b) rare class ($Y_i = +1$) | (c) common class ($Y_i = -1$) |

Figure 1: HIV-1 Protease Cleavage plots (2-standard error confidence bars). Comparison of misclassification test error rates among different regularizers.

We use the logistic loss $\ell(\theta; (x, y)) = \log(1 + \exp(-y\theta^\top x))$. We compare the performance of different constraint sets $\Theta$ by taking $\Theta = \left\{ \theta \in \mathbb{R}^d : a_1 \|\theta\|_1 + a_2 \|\theta\|_2 \leq r \right\}$, which is equivalent to elastic net regularization [27], while varying $a_1$, $a_2$, and $r$. We experiment with $\ell_1$-constraints ($a_1 = 1, a_2 = 0$) with $r \in \{50, 100, 500, 1000, 5000\}$, $\ell_2$-constraints ($a_1 = 0, a_2 = 1$) with $r \in \{5, 10, 50, 100, 500\}$, elastic net ($a_1 = 1, a_2 = 10$) with $r \in \{10^2, 2 \cdot 10^2, 10^3, 2 \cdot 10^3, 10^4\}$, our robust regularizer with $\rho \in \{10^2, 10^3, 10^4, 5 \cdot 10^4, 10^5\}$ and our robust regularizer coupled with the $\ell_1$-constraint ($a_1 = 1, a_2 = 0$) with $r = 100$. Though we use a convex surrogate (logistic loss), we measure performance of the classifiers using the zero-one (misclassification) loss $1\{\text{sign}(\theta^T x)y \leq 0\}$. For validation, we perform 50 experiments, where in each experiment we randomly select $9/10$ of the data to train the model, evaluating its performance on the held out $1/10$ fraction (test).

We plot results summarizing these experiments in Figure 1. The horizontal axis in each figure indexes our choice of regularization value (so "$\texttt{Regularizer = 1}$" for the $\ell_1$-constrained problem corresponds to $r = 50$). The figures show that the robustly regularized risk provides a different type of protection against overfitting than standard regularization or constraint techniques do: while other regularizers underperform in heavily constrained settings, the robustly regularized estimator $\widehat{\theta}_n^{\text{rob}}$ achieves low classification error for all values of $\rho$. Notably, even when coupled with a fairly stringent $\ell_1$-constraint ($r = 100$), robust regularization has performance better than $\ell_1$ except for large values $r$, especially on the rare label $Y = +1$.

We investigate the effects of the robust regularizer with a slightly different perspective in Table 1, where we use $\Theta = \{\theta : \|\theta\|_1 \leq 100\}$ for the constraint set for each experiment. We give error rates and logistic risk values for the different procedures, averaged over 50 independent runs. We note that all gaps are significant at the 3-standard error level. We see that the ERM solutions achieve good performance on the common class ($Y = -1$) but sacrifice performance on the uncommon class. As we increase $\rho$, performance of the robust solution $\widehat{\theta}_n^{\text{rob}}$ on the rarer label $Y = +1$ improves, while the error rate on the common class degrades a small (insignificant) amount.

Table 1: HIV-1 Cleavage Error

|         | risk   |        | error (%) |       | error ($Y = +1$) |       | error ($Y = -1$) |       |
|---------|--------|--------|-----------|-------|------------------|-------|------------------|-------|
| $\rho$  | train  | test   | train     | test  | train            | test  | train            | test  |
| erm     | 0.1587 | 0.1706 | 5.52      | 6.39  | 17.32            | 18.79 | 2.45             | 3.17  |
| 100     | 0.1623 | 0.1763 | 4.99      | 5.92  | 15.01            | 17.04 | 2.38             | 3.02  |
| 1000    | 0.1777 | 0.1944 | 4.5       | 5.92  | 13.35            | 16.33 | 2.19             | 3.2   |
| 10000   | 0.283  | 0.3031 | 2.39      | 5.67  | 7.18             | 14.65 | 1.15             | 3.32  |

## 4.2 Document classification in the Reuters corpus

For our second experiment, we consider a multi-label classification problem with a reasonably large dataset. The Reuters RCV1 Corpus [13] has 804,414 examples with $d = 47{,}236$ features, where feature $j$ is an indicator variable for whether word $j$ appears in a given document. The goal is to classify documents as a subset of the 4 categories where documents are labeled with a subset of those. As documents can belong to multiple categories, we fit binary classifiers on each of the four

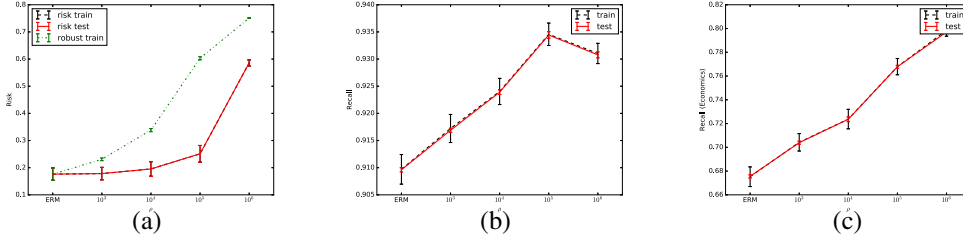

|     | (a) | (b) | (c) |

Figure 2: Reuters corpus experiment. (a) Logistic risks. (b) Recall. (c) Recall on Economics (rare).

categories. Each category has different number of documents (Corporate: $381, 327$, Economics: $119, 920$, Government: $239, 267$, Markets: $204, 820$) In this experiment, we expect the robust solution to outperform ERM on the rarer category (Economics), as the robustification (6) naturally upweights rarer (harder) instances, which disproportionally affect variance—as in the previous experiment.

For each category $k \in \{1, 2, 3, 4\}$, we use the logistic loss $\ell(\theta_k; (x, y)) = \log(1 + \exp(-y\theta_k^\top x))$. For each binary classifier, we use the $\ell_1$ constraint set $\Theta = \{\theta \in \mathbb{R}^d : \|\theta\|_1 \leq 1000\}$. To evaluate performance on this multi-label problem, we use precision (ratio of the number of correct positive labels to the number classified as positive) and recall (ratio of the number of correct positive labels to the number of actual positive labels). We partition the data into ten equally-sized sub-samples and perform ten validation experiments, where in each experiment we use one of the ten subsets for fitting the logistic models and the remaining nine partitions as a test set to evaluate performance.

In Figure 2, we summarize the results of our experiment averaged over the 10 runs, with 2-standard error bars (computed across the folds). To facilitate comparison across the document categories, we give exact values of these averages in Tables 2 and 3. Both $\widehat{\theta}_n^{\mathrm{rob}}$ and $\widehat{\theta}^{\mathrm{erm}}$ have reasonably high precision across all categories, with increasing $\rho$ giving a mild improvement in precision (from $.93 \pm .005$ to $.94 \pm .005$). On the other hand, we observe in Figure 2(c) that ERM has low recall ($.69$ on test) for the Economics category, which contains about 15% of documents. As we increase $\rho$ from 0 (ERM) to $10^5$, we see a smooth and substantial improvement in recall for this rarer category (without significant degradation in precision). This improvement in recall amounts to reducing variance in predictions on the rare class. This precision and recall improvement comes in spite of the increase in the average binary logistic risk for each of the 4 classes. In Figure 2(a), we plot the average binary logistic loss (on train and test sets) averaged over the 4 categories as well as the upper confidence bound $R_n(\theta, \mathcal{P}_n)$ as we vary $\rho$. The robust regularization effects reducing variance appear to improve the performance of the binary logistic loss as a surrogate for true error rate.

Table 2: Reuters Corpus Precision (%)

|       | Precision | | Corporate | | Economics | | Government | | Markets | |
| ----- | ----- | ----- | ----- | ----- | ----- | ----- | ----- | ----- | ----- | ----- |
| $\rho$ | train | test | train | test | train | test | train | test | train | test |
| erm | 92.72 | 92.7 | 93.55 | 93.55 | 89.02 | 89 | 94.1 | 94.12 | 92.88 | 92.94 |
| 1E3 | 92.97 | 92.95 | 93.31 | 93.33 | 87.84 | 87.81 | 93.73 | 93.76 | 92.56 | 92.62 |
| 1E4 | 93.45 | 93.45 | 93.58 | 93.61 | 87.6 | 87.58 | 93.77 | 93.8 | 92.71 | 92.75 |
| 1E5 | 94.17 | 94.16 | 94.18 | 94.19 | 86.55 | 86.56 | 94.07 | 94.09 | 93.16 | 93.24 |
| 1E6 | 91.2 | 91.19 | 92 | 92.02 | 74.81 | 74.8 | 91.19 | 91.25 | 89.98 | 90.18 |

Table 3: Reuters Corpus Recall (%)

|       | Recall | | Corporate | | Economics | | Government | | Markets | |
| ----- | ----- | ----- | ----- | ----- | ----- | ----- | ----- | ----- | ----- | ----- |
| $\rho$ | train | test | train | test | train | test | train | test | train | test |
| erm | 90.97 | 90.96 | 90.20 | 90.25 | 67.53 | 67.56 | 90.49 | 90.49 | 88.77 | 88.78 |
| 1E3 | 91.72 | 91.69 | 90.83 | 90.86 | 70.42 | 70.39 | 91.26 | 91.23 | 89.62 | 89.58 |
| 1E4 | 92.40 | 92.39 | 91.47 | 91.54 | 72.38 | 72.36 | 91.76 | 91.76 | 90.48 | 90.45 |
| 1E5 | 93.46 | 93.44 | 92.65 | 92.71 | 76.79 | 76.78 | 92.26 | 92.21 | 91.46 | 91.47 |
| 1E6 | 93.10 | 93.08 | 92.00 | 92.04 | 79.84 | 79.71 | 91.89 | 91.90 | 92.00 | 91.97 |

---

Code is available at `https://github.com/hsnamkoong/robustopt`.

**Acknowledgments**   We thank Feng Ruan for pointing out a much simpler proof of Theorem 1 than in our original paper. JCD and HN were partially supported by the SAIL-Toyota Center for AI Research and HN was partially supported Samsung Fellowship. JCD was also partially supported by the National Science Foundation award NSF-CAREER-1553086 and the Sloan Foundation.

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
