[Supplementary Material · nips-variance-regularization.pdf]

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

# A  Proofs of Main Results

In this section, we provide the proofs of all of our major results. Within each proof, we defer arguments for more technical and ancillary results to appendices as necessary.

## A.1  Proof of Theorem 1

Let $\sigma^2 = \mathrm{Var}(Z)$ and $s_n^2 = \mathrm{Var}_{\widehat{P}_n}(Z) = \mathbb{E}_{\widehat{P}_n}[Z^2] - \mathbb{E}_{\widehat{P}_n}[Z]^2$ denote the population and sample variance of $Z$, respectively. The theorem is immediate if $s_n = 0$ or $\sigma^2 = 0$, as in this case $\sup_{P:D_\phi(P\|\widehat{P}_n)\leq\rho/n} \mathbb{E}_P[Z] = \mathbb{E}_{\widehat{P}_n}[Z] = \mathbb{E}[Z]$. In what follows, we will thus assume that $\sigma^2, s_n^2 > 0$. Recall the maximization problem (7), which is

$$\underset{p}{\text{maximize}} \sum_{i=1}^n p_i z_i \text{ subject to } p \in \mathcal{P}_n = \left\{ p \in \mathbb{R}_+^n : \frac{1}{2} \|np - \mathbf{1}\|_2^2 \leq \rho, \langle \mathbf{1}, p \rangle = 1 \right\},$$

and the solution criterion (8), which guarantees that the maximizing value of problem (7) is $\overline{z} + \sqrt{2\rho s_n^2/n}$ whenever

$$\sqrt{2\rho} \frac{z_i - \overline{z}}{\sqrt{ns_n^2}} \geq -1.$$

Letting $z = Z$, then under the conditions of the theorem, we have $|z_i - \overline{z}| \leq M$, and to satisfy inequality (8) it is certainly sufficient that

$$2\rho \frac{M^2}{ns_n^2} \leq 1, \text{ or } n \geq \frac{2\rho M^2}{s_n^2}, \text{ or } s_n^2 \geq \frac{2\rho M^2}{n}. \tag{16}$$

Conversely, suppose that $s_n^2 < \frac{2\rho M^2}{n}$. Then we have $\frac{2\rho s_n^2}{n} < \frac{4\rho^2 M^2}{n^2}$, which in turn implies that

$$\sup_{p\in\mathcal{P}_n} \langle p, z \rangle \geq \frac{1}{n} \langle \mathbf{1}, z \rangle + \left( \sqrt{\frac{2\rho s_n^2}{n}} - \frac{2M\rho}{n} \right)_+.$$

Combining this inequality with the condition (16) for the exact expansion to hold yields the two-sided variance bounds (9).

We now turn to showing the high-probability exact expansion (10), which occurs whenever the sample variance is large enough by expression (16). To that end, we show that $s_n^2$ is bounded from below with high probability. Define the event

$$\mathcal{E}_n := \left\{ s_n^2 \geq \frac{1}{4}\sigma^2 \right\},$$

and let $n \geq \max\left\{ \frac{16\rho}{\sigma^2}, \frac{16}{\sigma^2}, 1 \right\} M^2$. Then, we have on event $\mathcal{E}_n$

$$n \geq \frac{16\rho M^2}{\sigma^2} \geq \frac{16}{4} \frac{\rho M^2}{s_n^2} = \frac{4\rho M^2}{s_n^2},$$

so that the sufficient condition (16) holds and expression (10) follows. We now argue that the event $\mathcal{E}_n$ has high probability via the following lemma, which is essentially an application of known concentration inequalities for convex functions coupled with a few careful estimates of the expectation of standard deviations.

**Lemma A.1.** *Let $Z_i$ be independent random variables taking values in $[M_0, M_1]$ with $M = M_1 - M_0$, and let $s_n^2 = \frac{1}{n}\sum_{i=1}^n Z_i^2 - \left( \frac{1}{n}\sum_{i=1}^n Z_i \right)^2$. For all $t \geq 0$, we have*

$$\mathbb{P}\left( s_n \geq \sqrt{\mathbb{E}s_n^2} + t \right) \vee \mathbb{P}\left( s_n \leq \sqrt{\mathbb{E}s_n^2} - \frac{M^2}{n} - t \right) \leq \exp\left( -\frac{nt^2}{2M^2} \right).$$

See Section B.2 for a proof of the lemma. When the $Z_i$ are i.i.d., we obtain

$$\mathbb{P}\left( s_n \geq \sigma\sqrt{1 - n^{-1}} + t \right) \vee \mathbb{P}\left( s_n \leq \sigma\sqrt{1 - n^{-1}}\sigma - \frac{M^2}{n} - t \right) \leq \exp\left( -\frac{nt^2}{2M^2} \right)$$

where $\sigma^2 = \mathrm{Var}(Z)$.

Now, substitute $t = \sigma(\sqrt{1 - n^{-1}} - \frac{1}{2}) - \frac{M^2}{n}$ so that

$$\sigma(1 - n^{-\frac{1}{2}}) - \frac{M^2}{n} - t = \frac{1}{2}\sigma.$$

Note that $\frac{M^2}{n} \leq \frac{M}{\sqrt{n}} \leq \sigma/4$ and since $\sqrt{1 - n^{-1}} \geq 1 - \frac{1}{2n\sqrt{1-n^{-1}}}$ and $\sigma^2 \leq M^2/4$ by standard variance bounds, our choice of $n$ also satisfies $n \geq 16M^2/\sigma^2 \geq 64$. We thus have $t/\sigma \geq 1 - \frac{1}{16\sqrt{63}} - \frac{1}{2} - \frac{1}{4} > \frac{1}{\sqrt{18}}$. We obtain

$$\mathbb{P}\left(\mathcal{E}_n\right) \geq 1 - \exp\left(-\frac{n\sigma^2\left((1 - n^{-1})^{1/2} - 1/2 - M^2/\sigma n\right)_+^2}{2M^2}\right) \geq 1 - \exp\left(-\frac{n\sigma^2}{36M^2}\right).$$

This gives the result (10).

## A.2 Proof of Theorem 2

Throughout this proof, we let $s_n^2(f) = \mathbb{E}_{\widehat{P}_n}[f(X)^2] - \mathbb{E}_{\widehat{P}_n}[f(X)]^2$ denote the empirical variance of the function $f$, and we use $\sigma_Q^2(f) = \mathbb{E}_Q[(f - \mathbb{E}_Q[f])^2]$ to denote the variance of $f$ under the distribution $Q$. Our starting point is to recall from inequality (16) in the proof of Theorem 1 that for each $f \in \mathcal{F}$, the empirical variance equality (11) holds if $n \geq \frac{4\rho M^2}{s_n^2(f)}$. As a consequence, Theorem 2 will follow if we can provide a uniform lower bound on the sample variances $s_n^2(f)$ that holds with high enough probability.

Set $\epsilon > 0$, and let $\{f_1, \ldots, f_N\}$, where $N = N(\mathcal{F}_{\geq \tau}, \epsilon, \|\cdot\|_{L^\infty(\mathcal{X})})$, be a minimal $\epsilon$-cover of $\mathcal{F}_{\geq \tau}$. Define the event

$$\mathcal{E}_n := \left\{s_n^2(f_i) \geq \frac{1}{4}\sigma^2(f_i) \text{ for } i = 1, \ldots, N\right\}.$$

By applying Lemma A.1 and the argument immediately following it in the proof of Theorem 1, we have

$$\mathbb{P}(\mathcal{E}_n) \geq 1 - \sum_{i=1}^N \exp\left(-\frac{n\sigma^2(f_i)\left(1/2 - 1/\sqrt{n} - M^2/\sigma(f_j)n\right)_+^2}{2M^2}\right)$$

$$\geq 1 - N(\mathcal{F}_{\geq \tau}, \epsilon, \|\cdot\|_{L^\infty(\mathcal{X})}) \exp\left(-\frac{n\tau^2\left(1/2 - 1/\sqrt{n} - M^2/\tau n\right)_+^2}{2M^2}\right). \qquad (17)$$

Thus, to obtain the theorem, we must show that the event $\mathcal{E}_n$ implies that the variance expansion (11) holds for each $g \in \mathcal{F}_{\geq \tau}$.

For $g \in \mathcal{F}$ and $f_j$ such that $\|g - f_j\|_{L^\infty(\mathcal{X})} \leq \epsilon$, the triangle inequality implies

$$|\sigma_Q(g) - \sigma_Q(f_j)| \leq \sqrt{\mathbb{E}_Q[(f_j - g + \mathbb{E}_Q g - \mathbb{E}_Q f_j)^2]}$$

$$= \sqrt{\mathbb{E}_Q[(f_j - g)^2] - (\mathbb{E}_Q[f_j - g])^2} \leq \epsilon$$

for any distribution $Q$ on $\mathcal{X}$. Then on the event $\mathcal{E}_n$, for any $g \in \mathcal{F}$ and the $f_j$ closest to $g$ from the covering, we have

$$s_n(g) \geq s_n(f_j) - \epsilon \geq \frac{1}{2}\sigma(f_j) - \epsilon \geq \frac{1}{2}\sigma(g) - \frac{3}{2}\epsilon.$$

That is, $s_n(g) \geq \frac{1}{2}\sigma(g) - \frac{3}{2}\epsilon$. Now, let $\epsilon = \frac{\tau}{24}$, which gives that

$$\mathcal{E}_n \text{ implies } s_n(g) \geq \frac{7}{16}\sigma(g) \geq \frac{7}{16}\tau \text{ for all } g \in \mathcal{F}_{\geq \tau}.$$

Recalling the sufficient condition (16) for the exact variance expansion to hold, we see that on $\mathcal{E}_n$,

$$\frac{4 \cdot 256\rho M^2}{49\tau^2} \geq \frac{4\rho M^2}{s_n^2(g)} \text{ for all } g \in \mathcal{F}_{\geq \tau}.$$

Taking $n \geq \frac{32\rho M^2}{\tau^2} > \frac{256\rho M^2}{49\tau^2}$ thus gives the result.

## A.3 Proof of Theorem 3

Before proving the theorem proper, we state two technical lemmas that provide uniform Bernstein-like bounds for the class $\mathcal{F}$. The first applies for empirical $\ell_\infty$-covering numbers.

**Lemma A.2** (Maurer and Pontil [16], Theorem 6). *Let $n \geq \frac{8M^2}{t}$ and $t \geq \log 12$. Then with probability at least $1 - 6N_\infty(\mathcal{F}, \epsilon, 2n)e^{-t}$, we have*

$$\mathbb{E}[f] \leq \mathbb{E}_{\widehat{P}_n}[f] + 3\sqrt{\frac{2\mathrm{Var}_{\widehat{P}_n}(f)t}{n}} + \frac{15Mt}{n} + 2\left(1 + 2\sqrt{\frac{2t}{n}}\right)\epsilon \tag{18}$$

*for all $f \in \mathcal{F}$.*

The second lemma applies when we have uniform $\|\cdot\|_{L^\infty(\mathcal{X})}$-covering numbers for $\mathcal{F}$.

**Lemma A.3.** *Let $\mathcal{F}$ be a collection functions with covering numbers $N(\mathcal{F}, \epsilon, \|\cdot\|_{L^\infty(\mathcal{X})})$, and assume that $|f(x)| \leq M$ for all $x$. Then with probability at least $1 - 2N(\mathcal{F}, \epsilon, \|\cdot\|_{L^\infty(\mathcal{X})})e^{-t}$,*

$$\mathbb{E}[f] \leq \mathbb{E}_{\widehat{P}_n}[f] + \sqrt{\frac{2\mathrm{Var}_{\widehat{P}_n}(f)t}{n-1}} + \frac{\sqrt{2t}M^2}{n^{3/2}-n} + \frac{2+3\sqrt{2}}{3}\frac{Mt}{n} + \left(2 + \sqrt{\frac{2t}{n-1}}\right)\epsilon$$

*and*

$$\mathbb{E}[f] \leq \mathbb{E}_{\widehat{P}_n}[f] + \sqrt{\frac{2\mathrm{Var}(f)t}{n}} + \frac{2Mt}{3n} + \left(2 + \sqrt{\frac{2t}{n-1}}\right)\epsilon$$

*for all $f \in \mathcal{F}$.*

As this lemma is essentially standard, we defer its proof to Section B.3.

We prove only the first set of bounds (13) in the theorem, which are based on Lemma A.2, as the proof of the bounds (12) follows in precisely the same way from Lemma A.3. We now return to the proof of Theorem 3. Let $\mathcal{E}_1$ denote that the event that the inequalities (18) hold. Now, let

$$\widehat{f} \in \underset{f \in \mathcal{F}}{\operatorname{argmin}} \sup_{P:D_\phi(P\|\widehat{P}_n) \leq \frac{\rho}{n}} \mathbb{E}_P[f(X)].$$

Then because $\mathcal{E}_1$ holds, we have

$$\mathbb{E}_P[\widehat{f}] \leq \mathbb{E}_{\widehat{P}_n}[\widehat{f}] + \sqrt{\frac{18\mathrm{Var}_{\widehat{P}_n}(\widehat{f}(X))t}{n}} + \frac{15Mt}{n} + 2\left(1 + 2\sqrt{\frac{2t}{n}}\right)\epsilon$$

$$\overset{(i)}{\leq} \sup_{P:D_\phi(P\|\widehat{P}_n) \leq \frac{\rho}{n}} \mathbb{E}_P[\widehat{f}(X)] + \sqrt{\frac{2\rho\mathrm{Var}_{\widehat{P}_n}(\widehat{f}(X))}{n}}$$

$$- \left(\sqrt{\frac{2\rho\mathrm{Var}_{\widehat{P}_n}(\widehat{f}(X))}{n}} - \frac{2M\rho}{n}\right)_+ + \frac{5M\rho}{3n} + 2\left(1 + 2\sqrt{\frac{2t}{n}}\right)\epsilon$$

$$\overset{(ii)}{\leq} \sup_{P:D_\phi(P\|\widehat{P}_n) \leq \frac{\rho}{n}} \mathbb{E}_P[f(X)] + \frac{11}{3}\frac{M\rho}{n} + 2\left(1 + 2\sqrt{\frac{2t}{n}}\right)\epsilon \text{ for all } f \in \mathcal{F}, \tag{19}$$

where inequality $(i)$ follows from the bounds (9) in Theorem 1 and the fact that $\rho \geq 9t$ by assumption and inequality $(ii)$ because $\widehat{f}$ minimizes $\sup_{P:D_\phi(P\|\widehat{P}_n) \leq \rho/n} \mathbb{E}_P[f(X)]$. This gives the first result (13a).

For the second result (13b), we bound the supremum term in expression (19). As the function $f$ is fixed, we have

$$\mathbb{E}_{\widehat{P}_n}[f] \leq \mathbb{E}[f] + \sqrt{\frac{2\mathrm{Var}(f)t}{n}} + \frac{2M}{3n}t$$

with probability at least $1 - e^{-t}$, and we similarly have by Lemma A.1 that

$$\sqrt{\mathrm{Var}_{\widehat{P}_n}(f)} \le \sqrt{1 - n^{-1}}\sqrt{\mathrm{Var}(f)} + \sqrt{\frac{2tM^2}{n}}$$

with probability at least $1 - e^{-t}$. That is, for any fixed $f \in \mathcal{F}$, we have with probability at least $1 - 2e^{-t}$ that

$$\sup_{P: D_\phi(P\|\widehat{P}_n) \le \frac{\rho}{n}} \mathbb{E}_P[f(X)] \overset{(i)}{\le} \mathbb{E}_{\widehat{P}_n}[f] + \sqrt{\frac{2\rho \mathrm{Var}_{\widehat{P}_n}(f)}{n}}$$

$$\le \mathbb{E}[f] + \sqrt{\frac{2\mathrm{Var}(f)t}{n}} + \frac{2M}{3n}t + \sqrt{\frac{2\rho\mathrm{Var}(f)}{n}} + \frac{2\sqrt{M^2\rho t}}{n}$$

$$\overset{(ii)}{\le} \mathbb{E}[f] + 2\sqrt{\frac{2\mathrm{Var}(f)\rho}{n}} + \frac{8}{3}\frac{M\rho}{n},$$

where inequality $(i)$ follows from the uniform upper bound (9) of Theorem 1 and inequality $(ii)$ from our assumption that $\rho \ge t$. Substituting this expression into our earlier bound (19) yields that for any $f \in \mathcal{F}$, with probability at least

$$1 - 2(3N_\infty(\mathcal{F}, \epsilon, 2n) + 1)e^{-t},$$

we have

$$\mathbb{E}[\widehat{f}(X)] \le \mathbb{E}[f(X)] + 2\sqrt{\frac{2\rho\mathrm{Var}(f(X))}{n}} + \frac{19}{3}\frac{M\rho}{n} + 2\left(1 + 2\sqrt{\frac{2t}{n}}\right)\epsilon.$$

This gives the theorem.

### A.4 Proof of Corollary 3.1

Let $\|f\|_{L^1(Q)} := \int |f(x)|dQ(x)$ denote the $L^1$-norm on $\mathcal{F}$ for the probability distribution $Q$. Then by Theorem 2.6.7 of van der Vaart and Wellner [24], we have

$$\sup_Q N(\mathcal{F}, \epsilon, \|\cdot\|_{L^1(Q)}) \le c\mathsf{VC}(\mathcal{F})\left(\frac{8Me}{\epsilon}\right)^{\mathsf{VC}(\mathcal{F})-1} \tag{20}$$

for a numerical constant $c$. Because $\|x\|_\infty \le \|x\|_1$, taking $Q$ to be uniform on $x \in \mathcal{X}^{2n}$ yields $N(\mathcal{F}(x), \epsilon, \|\cdot\|_\infty) \le N(\mathcal{F}, \frac{\epsilon}{2n}, \|\cdot\|_{L^1(Q)})$. The result follows by applying the bound (20) for $\epsilon/2n$.

### A.5 Proof of Corollary 3.3

.

Taking $\mathcal{F} = \{\ell(\theta, \cdot) : \theta \in \Theta\}$, any $\epsilon$-covering $\{\theta_1, \ldots, \theta_N\}$ of $\Theta$ in $\ell_2$-norm guarantees that $\min_i |\ell(\theta, x) - \ell(\theta_i, x)| \le L\epsilon$ for all $\theta, x$. That is,

$$N(\mathcal{F}, \epsilon, \|\cdot\|_{L^\infty(\mathcal{X})}) \le N(\Theta, \epsilon/L, \|\cdot\|_2) \le \left(1 + \frac{\mathrm{diam}(\Theta)L}{\epsilon}\right)^d,$$

where $\mathrm{diam}(\Theta) = \sup_{\theta, \theta' \in \Theta} \|\theta - \theta'\|_2$. Thus $\ell_2$-covering numbers of $\Theta$ control $L^\infty$-covering numbers of the family $\mathcal{F}$. Plugging in the respective values of $\rho$, $t$ and $\epsilon$ in Theorem 3, we obtain the result.

### A.6 Proof of Proposition 1

We certainly have $\ell(\theta^\star; x) = 0$ for all $x \in \mathcal{X}$, so that $\mathrm{Var}(\ell(\theta^\star; X)) = 0$. Now, consider the bound in Theorem 3 (12b). We see that $\log N(\{\ell(\theta, \cdot) : \theta \in \Theta\}, \epsilon, \|\cdot\|_{L^\infty(\mathcal{X})}) \le 2\log\frac{1}{\epsilon}$, and taking $\epsilon = \frac{1}{n}$, we have that if $\widehat{\theta}_n^{\mathrm{rob}} \in \operatorname{argmin}_{\theta \in \Theta} R_n(\theta, \mathcal{P}_n)$, then

$$R(\widehat{\theta}_n^{\mathrm{rob}}) \le R(\theta^\star) + \frac{15\rho}{n} \text{ with probability } \ge 1 - 4\exp(2\log n - \rho).$$

In particular, taking $\rho = 3 \log n$, we see that

$$R(\widehat{\theta}_n^{\text{rob}}) \le R(\theta^\star) + \frac{45 \log n}{n} \text{ with probability at least } 1 - \frac{4}{n}.$$

We now show the claim for th empirical risk minimizer. Let $\Phi(x) = \frac{1}{\sqrt{2\pi}} \int_{-\infty}^x e^{-\frac{1}{2}t^2} dt$ denotes the standard Gaussian CDF. (See Section B.1 for a proof.)

**Lemma A.4.** *Let the loss $\ell(\theta; x) = |\theta - x| - |x|$, $\delta \in [0, 1]$, and $X$ follow the distribution $P$ given by $P(X = 1) = \frac{1-\delta}{2}$, $P(X = -1) = \frac{1-\delta}{2}$, $P(X = 0) = \delta$.. Then with probability at least*

$$2\Phi\left(-\sqrt{\frac{n\delta^2}{1-\delta^2}}\right) - (1-\delta^2)^{\frac{n}{2}} \sqrt{\frac{8}{\pi n}},$$

*we have $R(\widehat{\theta}^{\text{erm}}) - R(\theta^\star) \ge \delta$.*

The risk for the empirical risk minimizer, as Lemma A.4 shows, may be substantially higher; taking $\delta = 1/\sqrt{n}$ we see that with probability at least $2\Phi(-\sqrt{\frac{n}{n-1}}) - 2\sqrt{2}/\sqrt{\pi e n} \ge 2\Phi(-\sqrt{\frac{n}{n-1}}) - n^{-\frac{1}{2}}$,

$$R(\widehat{\theta}^{\text{erm}}) \ge R(\theta^\star) + n^{-\frac{1}{2}}.$$

### A.7 Proof of Theorem 4

Recall our shorthand notation that $\pi(\theta) = \operatorname{argmin}_{\theta^* \in S}\{\|\theta - \theta^*\|_2\}$ denotes the Euclidean projection of $\theta$ onto $S$, which is a closed convex set. Define also the localized empirical deviation function

$$\Delta_n(\theta) := \mathbb{E}\left[\ell(\theta; X) - \ell(\pi(\theta); X)\right] - \mathbb{E}_{\widehat{P}_n}\left[\ell(\theta; X) - \ell(\pi(\theta); X)\right]. \tag{21}$$

We begin with the following

**Claim A.1.** *If $\widehat{S}^\epsilon \not\subset S^{2\epsilon}$, then*

$$\sup_{\theta \in S^{2\epsilon}} \left\{ \Delta_n(\theta) + \sqrt{\frac{2\rho}{n} \operatorname{Var}_{\widehat{P}_n}\left(\ell(\theta; X) - \ell(\pi(\theta); X)\right)} \right\} \ge \epsilon. \tag{22}$$

Deferring the proof of the claim, let us prove the theorem. First, the growth condition (14) shows that

$$S^{2\epsilon} \subset \left\{\theta \in \Theta : \|\theta - \pi(\theta)\|_2 \le \left(\frac{2\epsilon}{\lambda}\right)^{\frac{1}{\gamma}}\right\} = \left\{\theta \in \Theta : \operatorname{dist}(\theta, S) \le \left(\frac{2\epsilon}{\lambda}\right)^{\frac{1}{\gamma}}\right\}.$$

Therefore, we have for all $\theta \in S^{2\epsilon}$ that

$$\operatorname{Var}_{\widehat{P}_n}\left(\ell(\theta; X) - \ell(\pi(\theta); X)\right) \le L^2 \operatorname{dist}(\theta, S)^2 \le L^2 \left(\frac{2\epsilon}{\lambda}\right)^{\frac{2}{\gamma}},$$

and so by the assumption (15) that $\epsilon \ge \left(\frac{8L^2\rho}{n}\right)^{\frac{\gamma}{2(\gamma-1)}} \left(\frac{2}{\lambda}\right)^{\frac{1}{\gamma-1}}$, we have

$$\sqrt{\frac{2\rho}{n} \operatorname{Var}_{\widehat{P}_n}\left(\ell(\theta; X) - \ell(\pi(\theta); X)\right)} \le L\sqrt{\frac{2\rho}{n}} \left(\frac{2\epsilon}{\lambda}\right)^{\frac{1}{\gamma}} \le \frac{\epsilon}{2}.$$

In particular, if the event (22) holds then

$$\sup_{\theta \in S^{2\epsilon}} \Delta_n(\theta) \ge \frac{\epsilon}{2},$$

and recalling the definition (21) of $\Delta_n$, it then follows that

$$\mathbb{P}\left(\widehat{S}^\epsilon \not\subset S^{2\epsilon}\right) \le \mathbb{P}\left(\sup_{\theta \in S^{2\epsilon}} \Delta_n(\theta) \ge \frac{\epsilon}{2}\right). \tag{23}$$

To bound the probability (23), we use standard bounded difference and symmetrization arguments [e.g. 7, Theorem 6.5]. Letting $f(X_1, \ldots, X_n) := \sup_{\theta \in S^{2\epsilon}} \Delta_n(\theta)$, the function $f$ satisfies bounded differences:

$$\sup_{x,x' \in \mathcal{X}} |f(X_1, \cdots, X_{j-1}, x, X_{j+1}, \cdots, X_n) - f(X_1, \cdots, X_{j-1}, x', X_{j+1}, \cdots, X_n)|$$

$$\leq \sup_{x,x' \in \mathcal{X}} \sup_{\theta \in S^{2\epsilon}} \left| \frac{1}{n}(\ell(\theta; x) - \ell(\pi(\theta); x)) - \frac{1}{n}(\ell(\theta; x') - \ell(\pi(\theta); x')) \right|$$

$$\leq \frac{2L}{n} \sup_{\theta \in S^{2\epsilon}} \mathrm{dist}(\theta, S) \leq \frac{2L}{n} \left( \frac{2\epsilon}{\lambda} \right)^{\frac{1}{\gamma}}$$

for $j = 1, \ldots, n$. Using the standard symmetrization inequality $\mathbb{E}[\sup_{\theta \in S^{2\epsilon}} \Delta_n(\theta)] \leq 2\mathbb{E}[\mathfrak{R}_n(S^{2\epsilon})]$ and the bounded differences inequality [7, Theorem 6.5], we have

$$\mathbb{P}\left( \sup_{\theta \in S^{2\epsilon}} \Delta_n(\theta) \geq 2\mathbb{E}[\mathfrak{R}_n(S^{2\epsilon})] + t \right) \leq \exp\left( -\frac{nt^2}{2L^2} \left( \frac{\lambda}{2\epsilon} \right)^{\frac{2}{\gamma}} \right)$$

for all $t \geq 0$. Letting $u = \frac{nt^2}{2L^2} \left( \frac{\lambda}{2\epsilon} \right)^{\frac{2}{\gamma}}$ above and recalling the assumption (15) upper bounding $\mathbb{E}[\mathfrak{R}_n(S^{2\epsilon})]$, we have $\mathbb{P}(\sup_{\theta \in S^{2\epsilon}} \Delta_n(\theta) \geq \frac{\epsilon}{2}) \leq e^{-u}$. The first result of the theorem follows from the bound (23).

To show the last claim of Theorem 4, note that a minor extension of standard chaining arguments (see, for example, van der Vaart and Wellner [24, Section 2.2]), we have

$$\mathbb{E}[\mathfrak{R}_n(S^{2\epsilon})] \leq CL \left( \frac{2\epsilon}{\lambda} \right)^{\frac{1}{\gamma}} \sqrt{\frac{d}{n}}$$

for some numerical constant $C > 0$. Plugging this into the bound (15) and rearranging for $\epsilon$, we obtain the result.

**Proof of Claim A.1**   If $\widehat{S}^\epsilon \not\subset S^{2\epsilon}$, then certainly it is the case that there is some $\theta \in \Theta \setminus S^{2\epsilon}$ such that

$$R_n(\theta, \mathcal{P}_n) \leq \inf_{\theta \in \Theta} R_n(\theta, \mathcal{P}_n) + \epsilon \leq R_n(\pi(\theta), \mathcal{P}_n) + \epsilon.$$

Using the convexity of $R_n$, we have for all $t \in [0, 1]$ that

$$R_n(t\theta + (1-t)\pi(\theta), \mathcal{P}_n) \leq tR_n(\theta, \mathcal{P}_n) + (1-t)R_n(\pi(\theta), \mathcal{P}_n) \leq R_n(\pi(\theta), \mathcal{P}_n) + t\epsilon.$$

For all $t \in [0, 1]$, we have by definition of orthogonal projection (because the vector $\theta - \pi(\theta)$ belongs to the normal cone to $S$ and $\pi(\theta)$; cf. [11, Prop. III.5.3.3]) that $\pi(t\theta + (1-t)\pi(\theta)) = \pi(\theta)$. Thus, choosing $t$ appropriately there exists $\theta' \in \mathrm{bd}\, S^{2\epsilon}$ with $\theta' = t\theta + (1-t)\pi(\theta)$ and $\pi(\theta') = \pi(\theta)$, and $R_n(\theta', \mathcal{P}_n) \leq R_n(\pi(\theta'), \mathcal{P}_n) + \epsilon$.

Adding and subtracting the risk $R(\theta)$ and $R(\pi(\theta))$, we have that for some $\theta \in \mathrm{bd}\, S^{2\epsilon}$ that

$$R_n(\theta, \mathcal{P}_n) - R(\theta) + R(\pi(\theta)) - R_n(\pi(\theta)) \leq R(\pi(\theta)) - R(\theta) + \epsilon \leq -\epsilon,$$

where we have used that $R(\theta) = R(\pi(\theta)) + 2\epsilon$ by construction. Multiplying by $-1$ on each side of the preceding display and taking suprema, we find that

$$\epsilon \leq \sup_{\theta \in S^{2\epsilon}} \{R(\theta) - R_n(\theta, \mathcal{P}_n) - (R(\pi(\theta)) - R_n(\pi(\theta), \mathcal{P}_n))\}$$

$$\leq \sup_{\theta \in S^{2\epsilon}} \sup_{P: D_\phi(P \| \widehat{P}_n) \leq \rho/n} \{R(\theta) - R(\pi) + \mathbb{E}_P \left[ \ell(\pi(\theta); X) - \ell(\theta; X) \right]\}.$$

Applying the upper bound in inequality (9) of Theorem 1 gives the claim. $\qquad\square$

# B  Proofs of Technical Lemmas

## B.1  Proof of Lemma A.4

Defining $N_y := \mathrm{card}\{i \in [n] : X_i = y\}$ for $y \in \{-1, 0, 1\}$, we immediately obtain

$$\mathbb{E}_{\widehat{P}_n}[\ell(\theta; X)] = \frac{1}{n} \left[ N_{-1}|\theta + 1| + N_1|\theta - 1| + N_0|\theta| - (n - N_0) \right],$$

because $N_1 + N_{-1} + N_0 = n$. In particular, we find that the empirical risk minimizer $\theta$ satisfies

$$\widehat{\theta}^{\mathrm{erm}} := \operatorname*{argmin}_{\theta \in \mathbb{R}} \mathbb{E}_{\widehat{P}_n} [\ell(\theta; X)] = \begin{cases} 1 & \text{if } N_1 > N_0 + N_{-1} \\ -1 & \text{if } N_{-1} > N_0 + N_1 \\ \in [-1, 1] & \text{otherwise.} \end{cases}$$

On the events $N_1 > N_{-1} + N_0$ or $N_{-1} > N_0 + N_1$, which are disjoint, then, we have

$$R(\widehat{\theta}^{\mathrm{erm}}) = \delta = R(\theta^\star) + \delta.$$

Let us give a lower bound on the probability of this event. Noting that marginally $N_1 \sim \mathsf{Bin}(n, \frac{1-\delta}{2})$ and using $N_0 + N_{-1} = n - N_1$, we have $N_1 > N_0 + N_{-1}$ if and only if $N_1 > \frac{n}{2}$, and we would like to lower bound

$$\mathbb{P}\left(N_1 > \frac{n}{2}\right) = \mathbb{P}\left(\mathsf{Bin}\left(n, \frac{1-\delta}{2}\right) > \frac{n}{2}\right) = \mathbb{P}\left(\mathsf{Bin}\left(n, \frac{1+\delta}{2}\right) < \frac{n}{2}\right).$$

Letting $\Phi(t) = \frac{1}{\sqrt{2\pi}} \int_{-\infty}^t e^{-u^2/2} du$ denote the standard Gaussian CDF, then Zubkov and Serov [28] show that

$$\mathbb{P}\left(N_1 \geq \frac{n}{2}\right) \geq \Phi\left(-\sqrt{2n D_{\mathrm{kl}}\left(\frac{1}{2}\Big\|\frac{1+\delta}{2}\right)}\right)$$

where $D_{\mathrm{kl}}(p\|q) = p \log \frac{p}{q} + (1-p) \log \frac{1-p}{1-q}$ denotes the binary KL-divergence. We have by standard bounds on the KL-divergence [23, Lemma 2.7] that $D_{\mathrm{kl}}(\frac{1}{2}\|\frac{1+\delta}{2}) \leq \frac{\delta^2}{2(1-\delta^2)}$, so that

$$\mathbb{P}\left(N_1 > \frac{n}{2} \text{ or } N_{-1} > \frac{n}{2}\right) \geq 2\Phi\left(-\sqrt{\frac{n\delta^2}{1-\delta^2}}\right) - 2\mathbb{P}\left(N_1 = \frac{n}{2}\right).$$

For $n$ odd, the final probability is 0, while for $n$ even, we have

$$\mathbb{P}\left(N_1 = \frac{n}{2}\right) = 2^{-n} \binom{n}{n/2} (1-\delta^2)^{n/2} \leq (1-\delta^2)^{n/2} \sqrt{\frac{2}{\pi n}},$$

where the inequality uses that $\binom{2n}{n} \leq \frac{4^n}{\sqrt{\pi n}}$ by Stirling's approximation. Summarizing, we find that

$$\mathbb{P}\left(N_1 > \frac{n}{2} \text{ or } N_{-1} > \frac{n}{2}\right) \geq 2\Phi\left(-\sqrt{\frac{n\delta^2}{1-\delta^2}}\right) - (1-\delta^2)^{n/2} \sqrt{\frac{8}{\pi n}}.$$

## B.2 Proof of Lemma A.1

We use two technical lemmas in the proof of this lemma.

**Lemma B.1** (Samson [20], Corollary 3). *Let $f : \mathbb{R}^n \to \mathbb{R}$ be convex and $L$-Lipschitz continuous with respect to the $\ell_2$-norm over $[a, b]^n$, and let $Z_1, \ldots, Z_n$ be independent random variables on $[a, b]$. Then for all $t \geq 0$,*

$$\mathbb{P}(f(Z_{1:n}) \geq \mathbb{E}[f(Z_{1:n})] + t) \vee \mathbb{P}(f(Z_{1:n}) \leq \mathbb{E}[f(Z_{1:n})] - t) \leq \exp\left(-\frac{t^2}{2L^2(b-a)^2}\right).$$

**Lemma B.2.** *Let $Y_i$ be independent random variables with finite 4th moment. Then we have the following inequalities:*

$$\mathbb{E}\left[\left(\frac{1}{n}\sum_{i=1}^n Y_i^2\right)^{\frac{1}{2}}\right] \geq \left(\frac{1}{n}\sum_{i=1}^n \mathbb{E}[Y_i^2]\right)^{\frac{1}{2}} - \frac{1}{\sqrt{n}}\sqrt{\frac{\frac{1}{n}\sum_{i=1}^n \mathrm{Var}(Y_i^2)}{\frac{1}{n}\sum_{i=1}^n \mathbb{E}[Y_i^2]}} \tag{24a}$$

$$\mathbb{E}\left[\left(\frac{1}{n}\sum_{i=1}^n Y_i^2\right)^{\frac{1}{2}}\right] \geq \left(\frac{1}{n}\sum_{i=1}^n \mathbb{E}[Y_i^2]\right)^{\frac{1}{2}} - \frac{1}{n}\frac{\frac{1}{n}\sum_{i=1}^n \mathrm{Var}(Y_i^2)}{\frac{1}{n}\sum_{i=1}^n \mathbb{E}[Y_i^2]}, \tag{24b}$$

*and if $|Y_i| \leq C$ for with probability 1 for all $1 \leq i \leq n$, then*

$$\mathbb{E}\left[\left(\frac{1}{n}\sum_{i=1}^n Y_i^2\right)^{\frac{1}{2}}\right] \geq \left(\frac{1}{n}\sum_{i=1}^n \mathbb{E}[Y_i^2]\right)^{\frac{1}{2}} - \min\left\{\frac{C^2}{n}, \frac{C}{\sqrt{n}}\right\}. \tag{24c}$$

We defer the proof of Lemma B.2 to the end of this section in Section B.2.1.

The function $\mathbb{R}^n \ni z \mapsto \left\| (I - (1/n)\mathbf{1}\mathbf{1}^\top)z \right\|_2$ is 1-Lipschitz with respect to the Euclidean norm, so Lemma B.1 implies

$$\mathbb{P}\left(\sqrt{\mathrm{Var}_{\widehat{P}_n}(Z)} \geq \mathbb{E}[\sqrt{\mathrm{Var}_{\widehat{P}_n}(Z)}] + t\right) \vee \mathbb{P}\left(\sqrt{\mathrm{Var}_{\widehat{P}_n}(Z)} \leq \mathbb{E}[\sqrt{\mathrm{Var}_{\widehat{P}_n}(Z)}] - t\right) \leq \exp\left(-\frac{nt^2}{2M^2}\right).$$

Then

$$\mathbb{E}[\sqrt{\mathrm{Var}_{\widehat{P}_n}(Z)}] \leq \sqrt{\mathbb{E}[\mathrm{Var}_{\widehat{P}_n}(Z)]} = \sqrt{1 - n^{-1}}\sqrt{\mathrm{Var}(Z)},$$

giving the first inequality of the lemma. For the second, let $Y_i = Z_i - \frac{1}{n}\sum_{j=1}^n Z_j$ so that $s_n^2 = \frac{1}{n}\sum_{i=1}^n Y_i^2$. Applying Lemma B.2 with $C = M$, we obtain $\mathbb{E}[s_n] \geq \sqrt{\mathbb{E}[s_n^2]} - \frac{M^2}{n}$ which gives the result.

### B.2.1   Proof of Lemma B.2

We first prove the claim (24a). To see this, we use that

$$\inf_{\lambda \geq 0}\left\{\frac{a^2}{2\lambda} + \frac{\lambda}{2}\right\} = \sqrt{a^2} = |a|,$$

and taking derivatives yields that for all $\lambda' \geq 0$,

$$\frac{a^2}{2\lambda} + \frac{\lambda}{2} \geq \frac{a^2}{2\lambda'} + \frac{\lambda'}{2} - \left(\frac{a^2}{2\lambda'^2} - \frac{1}{2}\right)(\lambda - \lambda').$$

By setting $\lambda_n = \sqrt{\frac{1}{n}\sum_{i=1}^n Y_i^2}$, we thus have for any $\lambda \geq 0$ that

$$\mathbb{E}\left[\left(\frac{1}{n}\sum_{i=1}^n Y_i^2\right)^{\frac{1}{2}}\right] = \mathbb{E}\left[\frac{\sum_{i=1}^n Y_i^2}{2n\lambda_n} + \frac{\lambda_n}{2}\right]$$

$$\geq \mathbb{E}\left[\frac{\sum_{i=1}^n Y_i^2}{2n\lambda} + \frac{\lambda}{2}\right] + \mathbb{E}\left[\left(\frac{1}{2} - \frac{\sum_{i=1}^n Y_i^2}{2n\lambda^2}\right)(\lambda_n - \lambda)\right].$$

Now we take $\lambda = \sqrt{\frac{1}{n}\sum_{i=1}^n \mathbb{E}[Y_i^2]}$, and we apply the Cauchy-Schwarz inequality to obtain

$$\mathbb{E}\left[\left(\frac{1}{n}\sum_{i=1}^n Y_i^2\right)^{\frac{1}{2}}\right]$$

$$\geq \left(\frac{1}{n}\sum_{i=1}^n \mathbb{E}[Y_i^2]\right)^{\frac{1}{2}} - \frac{1}{2\lambda^2}\mathbb{E}\left[\left(\frac{1}{n}\sum_{i=1}^n(Y_i^2 - \mathbb{E}[Y_i^2])\right)^2\right]^{\frac{1}{2}}\mathbb{E}\left[\left(\left(\frac{1}{n}\sum_{i=1}^n Y_i^2\right)^{\frac{1}{2}} - \left(\frac{1}{n}\sum_{i=1}^n \mathbb{E}[Y_i^2]\right)^{\frac{1}{2}}\right)^2\right]^{\frac{1}{2}}$$

$$= \left(\frac{1}{n}\sum_{i=1}^n \mathbb{E}[Y_i^2]\right)^{\frac{1}{2}} - \frac{1}{2\sqrt{n}\lambda^2}\left(\frac{1}{n}\sum_{i=1}^n \mathrm{Var}(Y_i^2)\right)^{\frac{1}{2}}\mathbb{E}\left[\left(\left(\frac{1}{n}\sum_{i=1}^n Y_i^2\right)^{\frac{1}{2}} - \left(\frac{1}{n}\sum_{i=1}^n \mathbb{E}[Y_i^2]\right)^{\frac{1}{2}}\right)^2\right]^{\frac{1}{2}}$$

where the last equality follows from independence. Using the triangle inequality, we obtain that the final expectation is bounded by $2\lambda = 2\sqrt{\frac{1}{n}\sum_{i=1}^n \mathbb{E}[Y_i^2]}$, which gives inequality (24a). Now we give a sharper result. We have

$$\mathbb{E}\left[\left(\left(\frac{1}{n}\sum_{i=1}^n Y_i^2\right)^{\frac{1}{2}} - \left(\frac{1}{n}\sum_{i=1}^n \mathbb{E}[Y_i^2]\right)^{\frac{1}{2}}\right)^2\right]$$

$$= \frac{2}{n}\sum_{i=1}^n \mathbb{E}[Y_i^2] - 2\left(\frac{1}{n}\sum_{i=1}^n \mathbb{E}[Y_i^2]\right)^{\frac{1}{2}}\mathbb{E}\left[\left(\frac{1}{n}\sum_{i=1}^n Y_i^2\right)^{\frac{1}{2}}\right]$$

$$\leq \frac{2}{\sqrt{n}}\left(\frac{1}{n}\sum_{i=1}^n \mathrm{Var}(Y_i^2)\right)^{\frac{1}{2}}$$

where we used the first inequality (24a). Thus we also obtain the lower bound (24b). The final inequality follows immediately upon noticing that $\text{Var}(Y^2) \leq \mathbb{E}[Y^4] \leq C^2 \mathbb{E}[Y^2]$ for $\|Y\|_\infty \leq C$.

### B.3  Proof of Lemma A.3

By the standard Bernstein inequalities, we have that for any fixed $f$,

$$\mathbb{E}_{\widehat{P}_n}[f] \leq \mathbb{E}[f] + \sqrt{\frac{2\text{Var}(f)t}{n}} + \frac{2M}{3n}t \text{ with probability } \geq 1 - e^{-t}$$

$$\mathbb{E}_{\widehat{P}_n}[f] \geq \mathbb{E}[f] - \sqrt{\frac{2\text{Var}(f)t}{n}} - \frac{2M}{3n}t \text{ with probability } \geq 1 - e^{-t}.$$

By applying Lemma A.1 to upper bound $\text{Var}(f)$ with high probability (i.e. probability at least $1 - e^{-t}$), we then find that

$$\mathbb{E}[f] \leq \mathbb{E}_{\widehat{P}_n}[f] + \sqrt{\frac{2\text{Var}_{\widehat{P}_n}(f)t}{n-1}} + \frac{\sqrt{2t}M^2}{n^{3/2}-n} + \frac{2+3\sqrt{2}}{3}\frac{Mt}{n} \text{ with probability } \geq 1 - 2e^{-t}. \quad (25)$$

Now, let $\{f^1, \ldots, f^N\}$ be a minimal $\epsilon$-cover of $\mathcal{F}$ of size $N = N(\mathcal{F}, \epsilon, \|\cdot\|_{L^\infty(\mathcal{X})})$. Suppose that inequality (25) holds for each of the $f^i$. Then for any $f$ and $f^i$ satisfying $\|f - f^i\|_{L^\infty(\mathcal{X})} \leq \epsilon$, we have

$$\mathbb{E}[f] \leq \mathbb{E}[f^i] + \epsilon \leq \mathbb{E}_{\widehat{P}_n}[f^i] + \sqrt{\frac{2\text{Var}_{\widehat{P}_n}(f^i)t}{n-1}} + \frac{\sqrt{2t}M^2}{n^{3/2}-n} + \frac{2+3\sqrt{2}}{3}\frac{Mt}{n} + \epsilon$$

$$\leq \mathbb{E}_{\widehat{P}_n}[f^i] + \sqrt{\frac{2\text{Var}_{\widehat{P}_n}(f)t}{n-1}} + \frac{\sqrt{2t}M^2}{n^{3/2}-n} + \frac{2+3\sqrt{2}}{3}\frac{Mt}{n} + \left(1 + \sqrt{\frac{2t}{n-1}}\right)\epsilon,$$

where we have used that $\sqrt{\text{Var}(f^i)} = \sqrt{\text{Var}(f^i - f + f)} \leq \sqrt{\text{Var}(f^i - f)} + \sqrt{\text{Var}(f)}$ for any distribution, and $\text{Var}(f^i - f) \leq \|f^i - f\|^2_{L^\infty(\mathcal{X})} \leq \epsilon^2$. Noting that $\mathbb{E}_{\widehat{P}_n}[f^i] \leq \mathbb{E}_{\widehat{P}_n}[f] + \epsilon$ gives the result.

## C  Efficient solutions to computing the robust expectation

As a first step, we give a brief description of our (essentially standard) method for solving the robust risk problem. Our work in this paper focuses mainly on the properties of the robust objective (4) and its minimizers (6), so we only briefly describe the algorithm we use; we leave developing faster and more accurate specialized methods to further work. To solve the robust problem, we use a gradient descent-based procedure, and we focus on the case in which the empirical sampled losses $\{\ell(\theta, X_i)\}_{i=1}^n$ have non-zero variance for all parameters $\theta \in \Theta$, which is the case for all of our experiments.

Recall the definition of the subdifferential $\partial f(\theta) = \{g \in \mathbb{R}^d : f(\theta') \geq f(\theta) + \langle g, \theta' - \theta\rangle \text{ for all } \theta'\}$, which is simply the gradient for differentiable functions $f$. A standard result in convex analysis [11, Theorem VI.4.4.2] is that if the vector $p^* \in \mathbb{R}^n_+$ achieving the supremum in the definition (4) of the robust risk is unique, then

$$\partial_\theta R_n(\theta, \mathcal{P}_n) = \partial_\theta \sup_{P \in \mathcal{P}_n} \mathbb{E}_P[\ell(\theta; X)] = \sum_{i=1}^n p_i^* \partial_\theta \ell(\theta; X_i),$$

where the final summation is the standard Minkowski sum of sets. As this maximizing vector $p$ is indeed unique whenever $\text{Var}_{\widehat{P}_n}(\ell(\theta; X)) \neq 0$, we see that for all our problems, so long as $\ell$ is differentiable, so too is $R_n(\theta, \mathcal{P}_n)$ and

$$\nabla_\theta R_n(\theta, \mathcal{P}_n) = \sum_{i=1}^n p_i^* \nabla_\theta \ell(\theta; X_i) \text{ where } p^* = \operatorname*{argmax}_{p \in \mathcal{P}_n}\left\{\sum_{i=1}^n p_i \ell(\theta; X_i)\right\}. \quad (26)$$

In order to perform gradient descent on the risk $R_n(\theta, \mathcal{P}_n)$, then, by equation (26) we require only the computation of the worst-case distribution $p^*$. By taking the dual of the maximization (26), this

is an efficiently solvable convex problem; for completeness, we provide in the sequel a procedure for this computation that requires time $O(n \log n + \log \frac{1}{\epsilon} \log n)$ to compute an $\epsilon$-accurate solution to the maximization (26). As all our examples have smooth objectives, we perform gradient descent on the robust risk $R_n(\cdot, \mathcal{P}_n)$, with stepsizes chosen by a backtracking (Armijo) line search [8, Chapter 9.2].

We give a detailed description of the procedure we use to compute the supremum problem (7). In particular, our procedure requires time $O(n \log n + \log \frac{1}{\epsilon} \log n)$, where $\epsilon$ is the desired solution accuracy. Let us reformulate this as a minimization problem in a variable $p \in \mathbb{R}^n$ for simplicity. Then we wish to solve

$$\text{minimize } p^\top z \text{ subject to } \frac{1}{2n} \|np - \mathbf{1}\|_2^2 \le \rho, \ p \ge 0, \ p^\top \mathbf{1} = 1.$$

We take a partial dual of this minimization problem, then maximize this dual to find the optimizing $p$. Introducing the dual variable $\lambda \ge 0$ for the constraint that $\frac{1}{2} \|p - \frac{1}{n}\mathbf{1}\|_2^2 \le \frac{\rho}{n}$ and performing the standard min-max swap [8] (strong duality obtains for this problem because the Slater condition is satisfied by $p = \frac{1}{n}\mathbf{1}$) yields the maximization problem

$$\underset{\lambda \ge 0}{\text{maximize }} f(\lambda) := \inf_p \left\{ \frac{\lambda}{2} \left\|p - \frac{1}{n}\mathbf{1}\right\|_2^2 - \frac{\lambda \rho}{n} + p^\top z \mid p \ge 0, \ \mathbf{1}^\top p = 1 \right\}. \tag{27}$$

If we can efficiently compute the infimum (27), then it is possible to binary search over $\lambda$. Recall the standard fact [11, Chapter VI.4.4] that for a collection $\{f_p\}_{p \in \mathcal{P}}$ of concave functions, if the infimum $f(x) = \inf_{p \in \mathcal{P}} f_p(x)$ is attained at some $p_0$ then any vector $\nabla f_{p_0}(x)$ is a supergradient of $f(x)$. Thus, letting $p(\lambda)$ be the (unique) minimizing value of $p$ for any $\lambda > 0$, the objective (27) becomes $f(\lambda) = \frac{\lambda}{2} \|p(\lambda) - \frac{1}{n}\mathbf{1}\|_2^2 - \frac{\lambda \rho}{n} + p(\lambda)^\top z$, whose derivative with respect to $\lambda$ (holding $p$ fixed) is $f'(\lambda) = \frac{1}{2} \|p(\lambda) - \frac{1}{n}\mathbf{1}\|_2^2 - \frac{\rho}{n}$.

Now we use well-known results on the Euclidean projection of a vector to the probability simplex [9] to provide an efficient computation of the infimum (27). First, we assume with no loss of generality that $z_1 \le z_2 \le \cdots \le z_n$ and that $\mathbf{1}^\top z = 0$, because neither of these changes the original optimization problem (as $\mathbf{1}^\top p = 0$ and the objective is symmetric). Then we define the two vectors $s, \sigma^2 \in \mathbb{R}^n$, which we use for book-keeping in the algorithm, by

$$s_i = \sum_{j \le i} z_j, \ \ \sigma_i^2 = \sum_{j \le i} z_j^2,$$

and we let $z^2$ be the vector whose entries are $z_i^2$. The infimum problem (27) is equivalent to projecting the vector $v(\lambda) \in \mathbb{R}^n$ defined by

$$v_i = \frac{1}{n} - \frac{1}{\lambda} z_i$$

onto the probability simplex. Notably [9], the projection $p(\lambda)$ has the form $p_i(\lambda) = (v_i - \eta)_+$ for some $\eta \in \mathbb{R}$, where $\eta$ is chosen such that $\sum_{i=1}^n p_i(\lambda) = 1$. Finding such a value $\eta$ is equivalent [9, Figure 1] to finding the unique index $i$ such that

$$\sum_{j=1}^{i} (v_j - v_i) < 1 \ \text{ and } \ \sum_{j=1}^{i+1} (v_j - v_{i+1}) \ge 1,$$

taking $i = n$ if no such index exists (the sum $\sum_{j=1}^{i} (v_j - v_i)$ is increasing in $i$ and $v_1 - v_1 = 0$). Given the index $i$, algebraic manipulations show that $\eta = \frac{1}{n} - \frac{1}{i} - \frac{1}{i} \sum_{j=1}^{i} z_j / \lambda = \frac{1}{n} - \frac{1}{i} - \frac{1}{i} s_i / \lambda$ satisfies the equality $\sum_{i=1}^{n} (v_i - \eta)_+ = 1$ and that $v_j - \eta \ge 0$ for all $j \le i$ while $v_j - \eta \le 0$ for $j > i$. Of course, given the index $i$ and $\eta$, we may calculate the derivative $\frac{\partial}{\partial \lambda} f(\lambda)$ efficiently as well:

$$f'(\lambda) = \frac{\partial}{\partial \lambda} \left\{ \frac{\lambda}{2} \|p(\lambda) - n^{-1}\mathbf{1}\|_2^2 - \frac{\lambda \rho}{n} + p(\lambda)^\top z \right\}$$

$$= \frac{1}{2} \|p(\lambda) - n^{-1}\mathbf{1}\|_2^2 - \frac{\rho}{n} = \frac{1}{2} \sum_{j=1}^{i} (v_j - \eta - n^{-1})^2 + \frac{1}{2} \sum_{j=i+1}^{n} \frac{1}{n^2} - \frac{\rho}{n}$$

$$= \frac{1}{2} \sum_{j=1}^{i} \left( \frac{1}{\lambda} z_j + \eta \right)^2 + \frac{n-i}{2n^2} - \frac{\rho}{n} = \frac{\sigma_i^2}{2\lambda^2} + \frac{i\eta^2}{2} + \frac{s_i \eta}{\lambda} + \frac{n-i}{2n^2} - \frac{\rho}{n}.$$

> **Inputs:** Sorted vector $z \in \mathbb{R}^n$ with $\mathbf{1}^\top z = 0$, parameter $\rho > 0$, solution accuracy $\epsilon$
>
> ---
> SET $\lambda_{\min} = 0$ and $\lambda_{\max} = \lambda_\infty = \max\{n\,\|z\|_\infty, \sqrt{n/2\rho}\,\|z\|_2\}$
> SET $s_i = \sum_{j \le i} z_j$ and $\sigma_i^2 = \sum_{j \le i} z_j^2$
> WHILE $|\lambda_{\max} - \lambda_{\min}| > \epsilon \lambda_\infty$
>     SET $\lambda = \frac{\lambda_{\max} + \lambda_{\min}}{2}$
>     SET $(\eta, i) = \text{FINDSHIFT}(z, \lambda, s)$     // (Figure 4)
>     SET $f'(\lambda) = \frac{1}{2\lambda^2}\sigma_i^2 + \frac{\eta^2}{2}i^2 + \frac{\eta}{\lambda}s_i + \frac{n-i}{2n^2} - \frac{\rho}{n}$
>     IF $f'(\lambda) > 0$
>       SET $\lambda_{\min} = \lambda$
>     ELSE
>       SET $\lambda_{\max} = \lambda$
> SET $\lambda = \frac{1}{2}(\lambda_{\max} + \lambda_{\min})$, $(\eta, i) = \text{FINDSHIFT}(z, \lambda, s)$
> SET $p_i = \left(\frac{1}{n} - \frac{1}{\lambda}z_i - \eta\right)_+$ and RETURN $p$

Figure 3: Procedure FINDP to find the vector $p$ minimizing $\sum_{i=1}^n p_i z_i$ subject to the constraint $\frac{1}{2n}\|np - \mathbf{1}\|_2^2 \le \rho$. Method takes $\log\frac{1}{\epsilon}$ iterations of the loop.

Finding the index optimal $i$ can be done by a binary search, which requires $O(\log n)$ time, and $f'(\lambda)$ is then computable in $O(1)$ time using the vectors $s$ and $\sigma^2$. It is then possible to perform a binary search over $\lambda$ using $f'(\lambda)$, which which requires $\log\frac{1}{\epsilon}$ iterations to find $\lambda$ within accuracy $\epsilon$, from which it is easy to compute $p(\lambda)$ via $p_i(\lambda) = (v_i - \eta)_+ = \left(n^{-1} - \lambda^{-1}z_i - \eta\right)_+$.

We summarize this discussion with pseudo-code in Figures 3 and 4, which provide a main routine and sub-routine for finding the optimal vector $p$. These routines show that, once provided the sorted vector $z$ with $z_1 \le z_2 \le \cdots \le z_n$ (which requires $n \log n$ time to compute), we require only $O(\log\frac{1}{\epsilon} \cdot \log n)$ computations.

> **Inputs:** Sorted vector $z$ with $\mathbf{1}^\top z = 0$, $\lambda > 0$, vector $s$ with $s_i = \sum_{j \le i} z_j$
>
> ---
> SET $i_{\text{low}} = 1$, $i_{\text{high}} = n$
> IF $\frac{1}{n} - \frac{z_n}{\lambda} \ge 0$
>     RETURN $(\eta = 0, i = n)$
> WHILE $i_{\text{low}} \ne i_{\text{high}}$
>     $i = \frac{1}{2}(i_{\text{low}} + i_{\text{high}})$
>     $s_{\text{left}} = \frac{1}{\lambda}(i z_i - s_i)$       // (this is $s_{\text{left}} = \sum_{j=1}^i (v_j - v_i)$)
>     $s_{\text{right}} = \frac{1}{\lambda}((i+1)z_{i+1} - s_{i+1})$    // (this is $s_{\text{right}} = \sum_{j=1}^{i+1}(v_j - v_{i+1})$)
>     IF $s_{\text{right}} \ge 1$ AND $s_{\text{left}} < 1$
>       SET $\eta = \frac{1}{n} - \frac{1}{i} - \frac{1}{\lambda i}s_i$ and RETURN $(\eta, i)$
>     ELSE IF $s_{\text{left}} \ge 1$
>       SET $i_{\text{high}} = i - 1$
>     ELSE
>       SET $i_{\text{low}} = i + 1$
> SET $i = i_{\text{low}}$ and $\eta = \frac{1}{n} - \frac{1}{i} - \frac{1}{\lambda i}s_i$ and RETURN $(\eta, i)$

Figure 4: Procedure FINDSHIFT to find index $i$ and parameter $\eta$ such that, for the definition $v_i = \frac{1}{n} - \frac{1}{\lambda}z_i$, we have $v_j - \eta \ge 0$ for $j \le i$, $v_j - \eta \le 0$ for $j > i$, and $\sum_{j=1}^n (v_j - \eta)_+ = 1$. Method requires time $O(\log n)$.