[Reviews · NeurIPS 2017]

Reviewer 1



The article "Variance-based Regularization with Convex Objectives" considers the problem of the risk minimization in a wide class of parametric models. The authors propose the convex surrogate for variance which allows for the near-optimal and computationally efficient estimation. The idea of the paper is to substitute the variance term in the upper bound for the risk of estimation with the convex surrogate based on the certain robust penalty. The authors prove that this surrogate provides good approximation to the variance term and prove certain upper bounds on the overall performance of the method. Moreover, the particular example is provided where the proposed method beats empirical risk minimization. The experimental part of the paper considers the comparison of empirical risk minimization with proposed robust method on 2 classification problems. The results show the expected bahaviour, i.e. the proposed method is better than ERM on rare classes with not significant loss on more common classes. To sum up, I think that this is very strong work on the edge between the theoretical and practical machine learning. My only concern is that the paper includes involved theoretical analysis and is much more suitable for full journal publication than for short conference paper (the Arxiv paper is 47 pages long!!!).

Reviewer 2



This paper introduces a variance-based scheme for risk minimization, using ideas from distributionally optimization and Owen's empirical likelihood. The key concept is a robustly regularized risk able to capture both the bias and variance uniformly for all models in hypothesis spaces. It is shown that the minimization of this robust regularized risk automatically attains a nearly optimal balance between approximation and estimation errors. As compared to existing model selection approaches, a remarkable property of the approach here is that the resulting optimization problem is convex. Theoretical and empirical analysis are performed to show its advantage over ERM scheme. The paper is well written and well motivated. I only have some minor comments: (1) Definition 2.1 The first $\psi$ should be $\psi_n$ (2) Appendix, line 421, 426: it seems that you want to refer to (7) instead of (8)? (3) Appendix, equation below line 421, it seems that $E_p[\hat{f}(X)]$ should be $E_p[f(X)]$? The last term $\frac{2M\rho}{n}$ is missing in eq (22)? (4) Last two displays in the proof of Lemma A.3: it seems that $\mathcal{F}_l$ should be $\mathcal{F}_L$ and $\mathcal{F}$, respectively?

Reviewer 3



Summary This work offers an alternative to the traditional empirical risk minimization. The approximation of the bias and variance terms is often a non convex function in the parameter(s) to be estimated, which affects the computational efficiency of the procedure. Thus the authors propose an expansion of the variance (in the bias/variance decomposition of the risk) that enables them to write the robust empirical risk (on a certain class of probabilities) as the sum of the empirical risk and the variance term. It shows that the robust empirical risk plus a variance term is a good surrogate for the empirical risk. Thus whenever the loss function is convex, this approach provides a tractable convex formulation. Advantages & drawbacks The authors provide theoretical guarantees in high probability to ensure that robust empirical risk trades automatically between approximation and estimation errors. They provide asymptotic and non asymptotic result. Remarks It is not clear what is Owen's empirical likelihood. Can it be made more explicit ? The main important assumption is that the loss function has a compact support of the for $[-M,M]$ with $M\geq 1$. To derive the results of the paper, $M$ needs to be known. Is there any chance to extend your result to the regression setting nonetheless ?